# Segmenting Text and Learning Their Rewards for Improved RLHF in Language Models

## Abstract

Reinforcement learning from human feedback (RLHF) has been widely adopted to align language models (LMs) with human preference. Prior RLHF works typically take a bandit formulation, which, though intuitive, ignores the sequential nature of LM generation and can suffer from the sparse reward issue. While recent works propose dense token-level RLHF, treating each token as an action may be oversubtle to proper reward assignment. In this paper, we seek to get the best of both by training and utilizing a segment-level reward model, which assigns a reward to each semantically complete text segment that spans over a short sequence of tokens. For reward learning, our method allows dynamic text segmentation and compatibility with standard sequence-preference datasets. For effective RL-based LM training against segment reward, we generalize the classical scalar bandit reward normalizers into location-aware normalizer functions and interpolate the segment reward for further densification. With these designs, our method performs competitively on popular RLHF benchmarks in both reward modeling and LM policy learning. Ablation studies are conducted to further demonstrate our method.

## 1 Introduction

To align language models (LMs, *e.g.*, OpenAI, 2023; Reid et al., 2024) with human values, reinforcement learning (RL, Sutton & Barto, 2018) methods have been widely adopted to optimize the non-differentiable human preference, leading to the paradigm of reinforcement learning from human feedback (RLHF, Ouyang et al., 2022; Bai et al., 2022b). A prevailing approach in RLHF is to optimize the LMs by proximal policy optimization (PPO, Schulman et al., 2017) against a *bandit* reward model learned from human preference data, with KL regularization towards a pre-specified target distribution to avoid over-optimization on the reward model (Ziegler et al., 2019; Stiennon et al., 2020; Castricato et al., 2022). While this bandit approach is easier for reward modeling and has achieved remarkable success, language generation is intrinsically sequential, rather than simultaneous. Thus, from the view of optimizing human preference, assigning a bandit reward to entire text sequence induces the sparse reward (delayed feedback) issue (Andrychowicz et al., 2017; Marbach & Tsitsiklis, 2003), that often hurts RL-based LM training by increasing gradient variance and lowering sample efficiency (Takanobu et al., 2019; Wang et al., 2020; Guo et al., 2022; Snell et al., 2022).

As efforts to mitigate this sparse reward issue, prior works have developed methods to "ground" the sequence-level preference label into a dense token-level reward model (Yang et al., 2023; Zhong et al., 2024). While a dense per-token reward signal reduces the optimization complexity (Laidlaw et al., 2023), each action, however, is then defined as a single token, *i.e.*, a *sub-word* that is finer-grained than a word, especially with the BPE-style tokenizers (Gage, 1994; Sennrich et al., 2016). For instance, Llama 3.1's tokenizer (Dubey et al., 2024) has tokens as $\{\texttt{Brit},\texttt{ce},\texttt{cod},\texttt{neo},\texttt{redd},\dots\}$ that have less clear semantic meaning *per se* in any given context. The contribution of those tokens to the text sequence will inevitably depend on later tokens, making reward/credit assignment harder, especially under the prevailing RLHF paradigm of implementing the reward model as an off-the-shelf decoder-only transformer (*e.g.*, Ouyang et al., 2022; Bai et al., 2022b; Menick et al., 2022). Further, token-level reward implicitly assumes that the basic unit of a text sequence is *token*, which may not follow linguistics, where a more meaningful decomposition of text may be *phase* (including *word*) that can be more semantically complete and generally consists of a short sequence of tokens.

To retain the optimization benefit of dense reward for RLHF, while mitigating its potential reward assignment issue and linguistic counter-intuition, in this paper, we seek to train and utilize a *segment-level* reward model, which assigns a reward to each semantically meaningful segment of text sequence that constitutes a small amount of (or just one) tokens. With this construction, we define the action space in RLHF as "text segment," interpolating between the finest "per token" and the coarsest "full sequence" and potentially getting the benefit of both worlds: easier RL-based LM training owing to denser feedback and more accurate training guidance by the semantic completeness of each action.

Technically, we are motivated by prior works (Malinin & Gales, 2018; Li et al., 2024a) to implement a dynamic text sequence segmentation by thresholding the entropy of LM's predictive distributions, under the assumption that tokens within a semantically complete text segment can be more certainly predicted by prior tokens, while the start of a new segment is not (Wang et al., 2024b). To allow training the segment-level reward model by the standard sequence-preference labels via Bradley-Terry (BT, Bradley & Terry, 1952) loss, we differentiably aggregate segment rewards in a text sequence into a parametrized sequence evaluation. The learned segment-level reward model is then utilized in PPO-based policy learning, where we observe the unsuitability of classical reward normalizers, *i.e.*, the mean and standard deviation (std) of full sequence rewards. We address this issue by generalizing the classical bandit normalizers of scalar mean and std into a mean and a std function that output the reward normalizers at arbitrary location of the text sequence. In addition, we enhance PPO training by within-segment reward interpolation, which further densifies training signal and improves results.

We test our method on the performance of both the learned segment-level reward model and the subsequent PPO-trained LM policy. On popular RLHF benchmarks for reward modeling and LM policy learning, our method indicates competitive performance gain against both the classical bandit reward approach and recent token-level reward approach. We conduct a wide array of ablation studies to validate our design choices and provide further insight into our method.

## 2 MAIN METHOD

### 2.1 NOTATIONS AND BACKGROUND

In this section, we will define generic notations, provide background on the classical bandit RLHF, and then discuss RL formulation of LM generation underlying recent efforts on dense-reward RLHF.

**Generic Notations.** Both reward modeling and policy learning require text prompt $x$ and the corresponding response $y$. Reward model training turns the supervised fine-tuned model $\pi_{\text{SFT}}(\cdot \mid \cdot)$ (without the final unembedding layer) into a parametrized scalar-output model $r_\phi(\cdot, \cdot)$ with parameter $\phi$ that scores its input. The LM policy $\pi_\theta$, parametrized by $\theta$, is then optimized against $r_\phi$.

**Bandit Reward Model Training.** Reward model training assumes a dataset $\mathcal{D}_{\text{pref}} = \{(x, y^w, y^l)\}$ of prompt $x$ and the corresponding winning/chosen response $y^w$ and losing/rejected response $y^l$, where the label of winning/losing comes from human evaluation on the entire text sequence $y^w$ and $y^l$. In the classical bandit RLHF, reward model $r_\phi$ is trained by the binary classification BT loss

$$\mathcal{L}_{\text{bandit}}(\phi) = -\mathbb{E}_{(x, y^w, y^l) \sim \mathcal{D}_{\text{pref}}} \left[ \log \sigma \left( r_\phi(x, y^w) - r_\phi(x, y^l) \right) \right] , \qquad (1)$$

where $\sigma(u) = 1/(1 + \exp(-u))$ denotes the sigmoid function.

**PPO-based Bandit Policy Learning.** In policy learning, a set $\mathcal{D}_{\text{pol}} = \{x\}$ of text prompts $x$ is given. The LM policy $\pi_\theta$ is trained to generate outputs on $\mathcal{D}_{\text{pol}}$ optimizing the bandit reward from $r_\phi$, with a KL penalty towards $\pi_{\text{SFT}}$ to avoid reward over-optimization. Collectively, the objective is

$$\max_\theta \mathbb{E}_{x \sim \mathcal{D}_{\text{pol}}, y \sim \pi_\theta(\cdot \mid x)} \left[ r_\phi(x, y) - \beta \times \log \left( \pi_\theta(y \mid x) / \pi_{\text{SFT}}(y \mid x) \right) \right] , \qquad (2)$$

where $\beta$ is the KL coefficient. In practice, for PPO's training stability, the value of $r_\phi(x, y)$ is de-mean and de-std normalized based on statistics calculated on a calibration dataset, *e.g.*, $\mathcal{D}_{\text{pref}}$.

**RL Formulation of LM Generation.** By its sequential nature, LM generation is formulated as an Markov Decision Process (MDP) $\mathcal{M} = (\mathbb{S}, \mathbb{A}, P, \mathcal{R}, \gamma)$ (Sutton & Barto, 2018). Concretely, for state space $\mathbb{S}$, the state at timestep $t$, $s_t$, consists of the prompt $x$ and all generated tokens so far $a_{<t} =: [a_0, \ldots, a_{t-1}]$ with $a_{<0} =: \varnothing$, *i.e.*, $s_t =: [x, a_{<t}]$. $\mathbb{A}$ is the action space, where the action $a_t$ at step $t$ is a short-sequence/segment of tokens from the vocabulary in our segment-level reward/policy

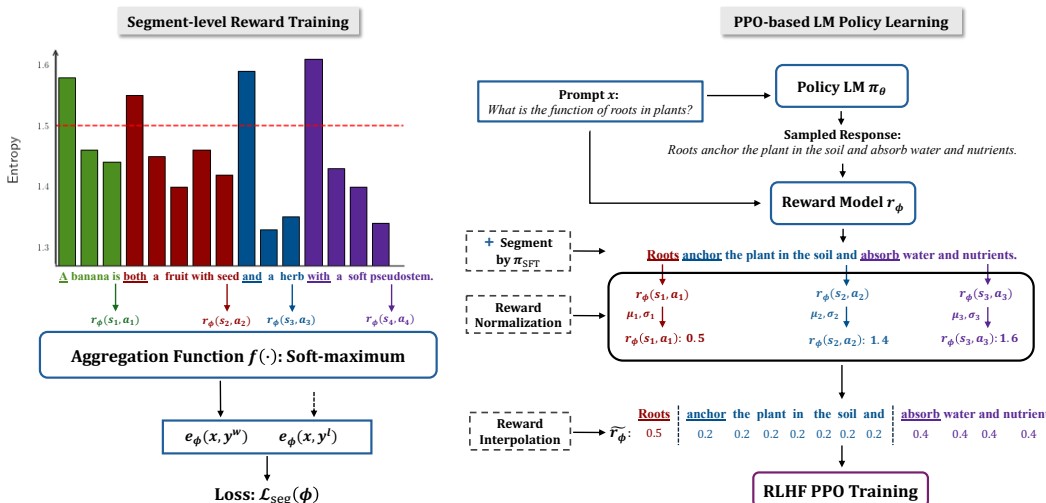

Figure 1: Overview of training and utilizing our segment-level reward model. Numerics in the plot are artificial. In the figure, each text segment has a different color, and its starting word is underscored.

setting while $a_t$ is a single token in the token-level reward/policy setting. Transition function $P$ deterministically appends the newly sampled tokens after the previous ones, *i.e.*, $s_{t+1} = [s_t, a_t] = [x, a_{\leq t}]$. $r(s, a) : \mathbb{S} \times \mathbb{A} \to \mathbb{R}$ scores the action choice (segment/token selection) $a$ at state/context $s$ and is typically substituted by the learned reward model $r_\phi$. $\gamma \in [0, 1]$ is the discount factor.

In what follows, we will focus on our segment-level reward/policy setting where each action $a_t \in \mathbb{A}$ is a semantically complete text segment, consisting of a non-deterministic number of consecutive tokens. The response $y$ for prompt $x$ then contains a variable number of segments/actions, generically denoted as $y = [a_0, \dots, a_{T-1}]$ where $T$ is the number of segments in $y$ and varies across responses. When necessary, we denote a single token in $y$ as $y_i$, whose generation context is $[x, y_{<i}]$.

Fig. 1 overviews key components in our method. A detailed algorithm box is deferred to Appendix A.

## 2.2 REWARD MODEL TRAINING

**Overview.** In training our segment-level reward model, we follow the data assumption set forth in Section 2.1, where the dataset $\mathcal{D}_{\text{pref}} = \{(x, y^w, y^l)\}$ contains only binary sequence-level preference labels, without any process supervision (Uesato et al., 2022). The reward model $r_\phi(s_t, a_t)$ is configured to output a scalar reward for each text segment choice $a_t$ at the generation context $s_t$. $r_\phi$ is trained such that its induced parameterized text sequence evaluations, aggregated from all segment-level rewards in the respective sequence, align with the preference labels in $\mathcal{D}_{\text{pref}}$. This is inspired by the imitation learning literature (*e.g.*, Christiano et al., 2017; Brown et al., 2019; 2020) and prior token-level reward modeling in RLHF (Yang et al., 2023). Collectively, the BT loss for training our segment-level reward function $r_\phi$ is

$$\mathcal{L}_{\text{seg}}(\phi) = -\mathbb{E}_{(x,y^w,y^l)\sim\mathcal{D}_{\text{pref}}}[\log \sigma \left(e_\phi(x, y^w) - e_\phi(x, y^l)\right)], \forall y \in \{y^w, y^l\}, e_\phi(x, y) = f(\{r_\phi(s_t, a_t)\}_{a_t \in y}), \quad (3)$$

where $e_\phi$ denotes the parameterized sequence evaluation induced by $r_\phi$, constructed by aggregating all segment-level rewards $\{r_\phi(s_t, a_t)\}_{a_t \in y}$ in the text sequence $y$ by a selected aggregation function $f(\cdot)$. Below, we discuss in details the segmentation method for text sequence and the choice of $f(\cdot)$.

**Entropy-based Segmentation.** As discussed in Section 1, we intend to split the given text sequence $y \in \{y^w, y^l\}$ into semantically complete segments, so that the reward assignment to each action (segment) can be easier, especially under the common implementation of the reward model as an auto-regressive LM. Recent works on LMs (*e.g.*, Li et al., 2024a; Wang et al., 2024b) have noticed that tokens within a semantically complete segment can be more predictable by the corresponding generation context, since they are continuation of the designated semantics; while the starting token of a new segment is comparably less predictable, as its semantic binding with prior words is relatively weaker. For auto-regressive LMs, the predictability of each token can be conveniently measured by the entropy of the next-token-prediction distribution from which the token is sampled (Malinin & Gales, 2018). To make text sequence segmentation a one-time data pre-processing in reward model

| **Prompt $x$:** Do artists/singers earn money from their songs being available on Spotify? How? | **Prompt $x$:** What species of trees tend to live the longest? |
|---|---|
| **Chosen Response $y^w$:**
Yes, they do! **Spotify pays them royalties for streaming**, which are **calculated as a percentage of their song plays.** Spotify **tracks a user's listening activity** (the number of times a user starts, pauses, and restarts a song) and this information is used to calculate the royalty. | **Chosen Response $y^w$:**
There's a lot of literature about this, and I think there's no consensus that all scientists agree on. But it's likely that the **bristlecone pines** in the **Sierra Nevada mountain range** would be the answer to your question. They are about **4,000 years** old and are **the oldest trees** on earth. |
| **Rejected Response $y^l$:**
Artists/singers are the owners of copyrights on their songs, and can use those copyrights to monetize their music however they choose. Spotify is a "Digital Service Provider" in the sense that it takes the rights to a song and acts as an intermediary to enable that song to be heard by users. | **Rejected Response $y^l$:**
There are a lot of variables which influence the length of a tree's life, including what sort of environment it lives in, the soil it grows in, and even its genes. In general, trees that are native to cold and dry climates tend to live longer than their counterparts in the tropics or southern hemisphere. |

Figure 2: Examples of data in our reward training dataset, motivating us to use soft-maximum as the aggregation function $f(\cdot)$. The key segments distinguishing the chosen response from the rejected one are highlighted in red color. We note that the choice of $f(\cdot)$ ought to be task and dataset specific, as discussed in Yang et al. (2023).

training, we choose to use the prediction distribution from the supervised fine-tuned model $\pi_{\mathrm{SFT}}$, from which the reward model is initialized before training. With a selected entropy cutoff $c_{\mathrm{ent}}$, token $y_i$ starts a new segment if the Shannon entropy $\mathcal{H}(\cdot)$ of $\pi_{\mathrm{SFT}}$'s predictive distribution of the $i$-th token surpasses $c_{\mathrm{ent}}$, *i.e.*, $\mathcal{H}(\pi_{\mathrm{SFT}}(\cdot \mid x, y_{<i})) > c_{\mathrm{ent}}$, in which case $y_{i-1}$ ends the previous segment.

**Choice of the Aggregation Function $f(\cdot)$.** Aggregation function $f(\cdot)$ provides inductive bias on the relation between the quality of each segment/action and the preferability of overall text sequence. Since $f$ probes into what kind of text sequences will be preferred, its selection should ideally be task and dataset specific, to avoid mis-specification and the subsequent unintended bias in reward model training. While several designs have been proposed in literature (Christiano et al., 2017; Kim et al., 2023; Yang et al., 2023), after looking into the dataset, in our experiments, we select soft-maximum to differentiably highlight the contribution of key segments. With this choice of $f(\cdot)$, the parametrized sequence evaluation $e_\phi(x, y)$ in Eq. (3) is constructed as

$$e_\phi(x, y) = f(\{r_\phi(s_t, a_t)\}_{a_t \in y}) = \tau \times \log\left[\sum_{a_t \in y} \exp(r_\phi(s_t, a_t)/\tau)\right], \quad (4)$$

where $\tau$ is the temperature controlling the sharpness of the log-sum-exp function. Fig. 2 provides example data-points from the reward training dataset in our experiments (Section 4), which support our choice of soft-maximum as the aggregation $f(\cdot)$. Other datasets may require a different $f(\cdot)$.

**An Alternative Interpretation.** Comparing our segment-level reward training loss Eq. (3) with the classical bandit loss Eq. (1), one may alternatively interpret $e_\phi$ and $f(\{r_\phi\})$ in Eq. (3) as a re-parametrization of the learned sequence-level feedback that differentiably aggregates the quality/-contribution of each text segment, and thereby connects a denser evaluation $r_\phi$ of each semantically complete text segment with the information in ground-truth sequence-level preference label.

### 2.3 PPO-BASED POLICY LEARNING

**Overview.** In policy learning, we again follow the classical bandit setting in Section 2.1 to optimize the LM policy $\pi_\theta$ on a given prompt set $\mathcal{D}_{\mathrm{pol}} = \{x\}$. But unlike the bandit objective in Eq. (2), we adopt the full RL setting (Sutton & Barto, 2018) to maximize $\pi_\theta$'s expected sum of per-segment/step rewards. This enables directly plugging our segment-level reward model $r_\phi$ into most off-the-shelf RLHF PPO implementation. With this, the policy learning objective for $\pi_\theta$ is

$$\max_\theta \mathbb{E}_{x \sim \mathcal{D}_{\mathrm{pol}}, y \sim \prod_{t=0}^{T-1} \pi_\theta(a_t \mid s_t)} \left[\sum_{t=0}^{T-1} r_\phi(s_t, a_t) - \beta \times \log\left(\pi_\theta(y \mid x)/\pi_{\mathrm{SFT}}(y \mid x)\right)\right], \quad (5)$$

where again, each $a_t$ is a segment of tokens (chopped by $\pi_{\mathrm{SFT}}$), $s_t = [x, a_0, \ldots a_{t-1}]$ is the generation context at step $t$, and $y = [a_0, \ldots, a_{T-1}]$ is the response to prompt $x$ sampled from the learning $\pi_\theta$.

Recall from Section 2.1 that the output values from the reward model $r_\phi$ need to be normalized for the stability of PPO training. With our segment-level reward model, it is no longer suitable to normalize each $r_\phi(s_t, a_t)$ by the mean and std of entire sequences' rewards as in the bandit setting, since the latter may not be on a proper scale and/or well-defined. Further, the on-policy nature of PPO induces

an extra complexity: each step of PPO samples new text sequences, whose length, segment lengths, and segment locations are all stochastic and can differ from the reward calibration dataset, *e.g.*, $\mathcal{D}_{\text{pref}}$. Below, we discuss our approach to construct the reward value normalizers, followed by interpolating the segment-level reward into per-token signal to helpfully provide an even denser training guidance.

**Location-aware Reward Normalizers via Regression.** While the length of the sampled response $y$ and the lengths and locations of segments $\{a_t\}$ in $y$ are all uncontrollable, we know that each $a_t$ is somewhere in $y$. Correspondingly, each input $(s_t, a_t)$ to $r_\phi$ is linked to a location indicator $p \in (0, 1]$ of $y$, and $p$ can be simplest defined as $t/T$, where $t$ is the index of the segment $a_t$ in $y$, since PPO routine has fully sampled $y$. On each datapoint in the calibration set, location indicator $p \in (0, 1]$ again, with the linked segment-level reward available. Across all datapoints in the calibration set, we construct a new dataset $\mathcal{D}_{\text{norm}} = \{(p, \mu_p, \sigma_p)\}$, where $p$ runs over all values of location indicator in the calibration set, $\mu_p$ and $\sigma_p$ respectively denote sample mean and sample std of all segment-level rewards corresponding to $p$ in the calibration set. With $\mathcal{D}_{\text{norm}}$, we run a simple linear regression to estimate the relation between location indicator $p$ and mean/std of segment-level rewards at $p$, *i.e.*,

$$\text{Mean}(p) = w_\mu \times p + b_\mu, \ \log \text{Std}(p) = w_\sigma \times p + b_\sigma \rightarrow \text{Std}(p) = \exp(w_\sigma \times p + b_\sigma), \quad (6)$$

where regression coefficients $(w_\mu, b_\mu), (w_\sigma, b_\sigma)$ can be calculated in closed form.

Note that the classical bandit normalizers of the mean and std of full sequences' rewards correspond to evaluate $\text{Mean}(p)$ and $\text{Std}(p)$ at $p = 1.0$. In this regard, our mean and std functions in Eq. (6) generalize the classical scalar normalizers into location-aware functions able to output proper reward normalizers at an arbitrary location $p$ of the text sequence. With $\text{Mean}(\cdot)$ and $\text{Std}(\cdot)$ and the corresponding $p$, $r_\phi(s_t, a_t)$ is normalized by $r_\phi(s_t, a_t) \leftarrow (r_\phi(s_t, a_t) - \text{Mean}(p))/\text{Std}(p)$.

**Within-segment Reward Interpolation.** Depending on the specific tokenizer in use, we observed that semantically complete text segments may contain up to around twenty tokens. The corresponding action space $\mathbb{A}$ might still be large and the resulting segment-level design might not sufficiently address the sample inefficiency issue in the classical bandit RLHF and could again lead to inferior PPO-based RL training. To further densify the RL training signal, we evenly split the segment-level reward $r_\phi(s_t, a_t)$ for a segment $a_t$ to each token $y_i \in a_t$. This induces a token-level credit assignment that $\forall y_i \in a_t, \widetilde{r}_\phi([x, y_{<i}], y_i) = r_\phi(s_t, a_t)/|a_t|$, where $[x, y_{<i}]$ is the generation context of token $y_i$. $\widetilde{r}_\phi$ can then directly substitute $r_\phi$ in Eq. (5), since $\sum_{t=0}^{T-1} r_\phi(s_t, a_t) = \sum_{t=0}^{T-1}(\sum_{y_i \in a_t} r_\phi(s_t, a_t)/|a_t|)$.

Note that $\widetilde{r}_\phi$ is still intrinsically segment level, since all token selections $y_i$ within segment $a_t$ receive the same feedback, *i.e.*, the average of segment-level reward $r_\phi(s_t, a_t)$. This is in contrast to prior works on token-level reward models (Yang et al., 2023; Zhong et al., 2024), where each token selection is evaluated separately and thus their token-level feedback/rewards vary across each token.

**Summary.** With the learned segment-level reward model $r_\phi$ from Section 2.2, in PPO training of the LM policy $\pi_\theta$, we first normalize each $r_\phi(s_t, a_t)$ in the sampled sequence by the corresponding normalizers $\text{Mean}(p)$ and $\text{Std}(p)$. Normalized segment-level rewards are then interpolated into per-token feedback signal $\widetilde{r}_\phi$. Finally, we plug $\widetilde{r}_\phi$ directly into an off-the-shelf RLHF PPO routine.

## 3 RELATED WORK

**Reward Models in RLHF.** In the classical RLHF paradigm, policy LM is optimized against a bandit reward model trained firstly by binary classification loss on the preference dataset, with KL penalty to a specified prior distribution to avoid reward over-optimization (Ziegler et al., 2019; Stiennon et al., 2020; Jaques et al., 2020; Bai et al., 2022a; Ouyang et al., 2022; Castricato et al., 2022). Under the same bandit formulation, recent works have enhanced the bandit reward model by directly modeling the probability of one response being preferred over the other (Jiang et al., 2023; Zhao et al., 2023; Liu et al., 2023; Dong et al., 2024) or factorizing human preference into multiple facets via multi-objective modeling (Touvron et al., 2023; Wang et al., 2023; 2024c;a). Despite its popularity, from the angle of RL-based optimization of human preference captured by the reward model, such a bandit reward may lead to inferior training, due to the sparse reward issue intrinsic to the bandit formulation of LM generation and credit assignment (*e.g.*, Takanobu et al., 2019; Guo et al., 2022).

Viewing the weakness of bandit RLHF, efforts have been making to densify the reward signal for RLHF LM training. Yang et al. (2023) and Chan et al. (2024) train token-level reward models by the binary preference classification loss. Zhong et al. (2024) and Rafailov et al. (2024) use an LM

trained by DPO (Rafailov et al., 2023) firstly for token-level reward assignment, which is later used in PPO training or search-based algorithms. Guo et al. (2023), Cao et al. (2024), and Yoon et al. (2024) assign continuous or fixed fine-grained rewards (*e.g.*, $\pm 1$) by accessing an external powerful large LM or the oracle environmental reward; while Chen et al. (2024) require the extra task and datasets of erroneous solution rewriting. Apart from potential extra requirements, as discussed in Section 1, the semantic incompleteness of *token* in text may challenge the efficacy of per-token credit assignment, especially with the prevailing implementation of reward model as a decoder-only transformer that cannot look ahead into later tokens. In contrast, by defining *text segment* as the basic unit of text sequence that can be semantically more complete than *token*, our segment-level reward may provide more accurate guidance for RL-based LM training, while not losing the benefit of denser feedback.

Close to our segment-level reward, process reward models (PRMs, *e.g.*, Uesato et al., 2022; Lightman et al., 2023) in reasoning-alike tasks also assign a single reward value to a short sequence of tokens. The training of PRMs, however, typically requires human annotation on each step of the reasoning-alike process. This may not be feasible in general text generation tasks, *e.g.*, text summarization or dialog, where each step/text segment lacks clear human evaluation criterion while the full generations can be more easily compared or evaluated. By contrast, as seen in Section 2, our method is developed for the most basic yet general RLHF setting, where human preference is only manifested in a dataset of binary sequence-level preference. And the dataset is collected from multiple sources and contains multiple forms of prompt-responses. We discuss a broader set of related works in Appendix D.

## 4 EXPERIMENTS

### 4.1 EXPERIMENTAL SETUPS AND IMPLEMENTATION

**Datasets.** For reward model training, we use the preference-700K dataset[1], which is a diverse collection of open-source preference datasets, such as HH-RLHF (Bai et al., 2022a), Stanford Human Preferences Dataset (SHP) (Ethayarajh et al., 2022), and HelpSteer (Wang et al., 2023). PPO-based LM policy training is performed on the Ultrafeedback dataset (Cui et al., 2023), from which we only use the prompts to sample responses during the PPO training routine.

**Benchmarks and Evaluations.** We evaluate the reward model performance on the RewardBench benchmark (Lambert et al., 2024). Each test sample consists of a triplet of a prompt, a chosen response, and a rejected response. The evaluation metric is the classification accuracy of chosen/rejected label. For our segment-level reward model, the parameterized sequence evaluation $e_\phi$ in Eq. (4) is used as the classification logit, and similarly for other dense reward models such as the token-level model.

The (PPO-trained) LM policy is evaluated on three popular open-ended instruction-following benchmarks: AlpacaEval 2.0 (Li et al., 2023), Arena-Hard (Li et al., 2024c), and MT-Bench (Zheng et al., 2023), where GPT-4o is used as the judge. Our reported scores follow each benchmark's default protocol. AlpacaEval 2.0 consists of 805 test cases, on which we report the length control win rate (`LC`), raw win rate (`WR`), and the average response length in number of characters (`# char`). The `LC` metric is specifically designed to be robust against model verbosity. We follow the convention (*e.g.*, Dong et al., 2024) to employ `alpaca_eval_gpt4_turbo_fn` as the annotator for AlpacaEval 2.0. Arena-Hard consists of 500 challenging user queries, and we report both the win rate (`WR`) against the reference model GPT-4-0314 and the average number of tokens in the responses (`# token`). MT-Bench is a multi-turn dialogue dataset that includes 80 test cases. We report the average MT-Bench scores for two dialogue turns, using GPT-4o as the judge model.

**Implementation.** Due to our limited compute resources, we currently implement our method onto the 3.8B SFT checkpoint of Phi3.1-mini Instruct (Abdin et al., 2024), which we use as the starting points of both reward model training and PPO-based LM policy learning, in the latter initializing the models for value function, learning policy, and reference policy. Our implementation is built upon the open-source RLHF framework OpenRLHF (Hu et al., 2024). We maximally follow the default hyperparameters in OpenRLHF. For example, in PPO training, we retain the default KL coefficient value $\beta = 0.01$. In both reward model training and LM policy learning, we train the models for one epoch, *i.e.*, one pass through the dataset, using entropy cutoff $c_{\text{ent}} = 2.0$ and temperature $\tau = 0.5$ in the soft-maximum aggregation (Eq. (4)). Section 4.3 presents ablation studies on the choice of

---

[1] https://huggingface.co/datasets/hendrydong/preference_700K

Table 1: Performance comparison among different action definitions, comparing both the resulted reward model and PPO-trained LM policy. "Avg Acc." is the average accuracy over the entire RewardBench. # {char, token} measures the average response length in the benchmark tests. Highest value of each column is in bold.

| Action Definition | RewardBench Avg Acc.(%) | AlpacaEval 2.0 | | | Arena-Hard | | MT-Bench |
|---|---|---|---|---|---|---|---|
| | | LC(%) | WR(%) | # char | WR(%) | # token | GPT-4o |
| SFT (No RLHF) | - | 14.93 | 10.19 | 1271 | 14.5 | 476 | 7.00 |
| Bandit (Sequence) | 81.11 | 14.98 | 12.05 | 1520 | 17.8 | 496 | 7.18 |
| Sentence | 84.42 | 15.79 | **16.52** | **2237** | 18.5 | **617** | 6.97 |
| Token | 81.20 | 17.00 | 14.78 | 1711 | 19.0 | 533 | 7.24 |
| Segment (**Ours**) | **85.93** | **19.68** | 16.15 | 1622 | **20.0** | 518 | **7.31** |
| Bandit as Segment | - | 13.86 | 9.60 | 1331 | 13.9 | 444 | 7.27 |
| Segment as Bandit | - | 15.58 | 13.29 | 1652 | 19.1 | 504 | 7.23 |

$c_{ent}$, $\tau$, and aggregation function. Due to space limit, we defer further implementation details to Appendix C. For reproducibility, our source code and model checkpoints are anonymously released.

## 4.2 MAIN EXPERIMENTAL COMPARISONS

**Baselines.** To demonstrate our unique consideration on RLHF's action space, in the main experiment, we compare our design of segment-level action space with the coarsest bandit/sequence-level action space, the coarser sentence-level space, and the finest token-level space, in terms of the performance of both the reward model and the subsequent PPO-trained LM policy. Unless specified, both the reward model and the LM policy are trained under the same action definition. The sentence-level models are implemented by splitting the text sequences by the sentence splitters {`"."`, `"!"`, `"?"`, `"\n"`, `";"`, `"..."`, `","`, `":"`} and/or their foreign language equivalents.

To further illustrate our segment-level reward model and denser segment-level reward assignment, we additionally compare with two hybrid approaches: *(A)* using the bandit reward model for segment-level reward assignment in the PPO training ("Bandit as Segment"); and *(B)* using the segment-level reward model only for bandit reward assignment in the PPO training ("Segment as Bandit"), where the bandit reward is implemented by the parametrized sequence evaluation $e_\phi$ in Eq. (4). For all baselines, we follow the standard training receipts and tune them to the extent of ensuring fair comparison.

**Results.** Table 1 compares the performance of our segment-level reward model and the resulted PPO-trained LM policy with those from other definitions of RLHF's action space, as well as the two hybrid approaches. The break-down scores of each reward model on each of the four categories in RewardBench is deferred to Table 5 in Appendix B.1. Our key findings are summarized as follows.

*(1) Segment-level action space improves reward modeling.* From Table 1's RewardBench results, it is clear that our segment-level reward model outperforms reward models from alternative RLHF action definitions, which we attribute to our design of semantically complete text sequence segmentation and a targeted choice of aggregation function in reward model training. This is corroborated by the strong result of sentence-level reward model, which performs a comparatively coarser text segmentation but otherwise also employs soft-maximum aggregation to highlight key sentences. Since it is coarser than our segment-level approach, it may not accurately pinpoint the contribution of most important phases/words, leading to its under-performance to ours. At one end of granularity spectrum, without finer reward/credit assignment and key phase highlight, the coarsest classical bandit/sequence-level reward model performs much weaker than both segment- and sentence-level models. On the other extreme, the recent finest token-level reward model, which ignores the semantic completeness in the action space definition, suffers from accurate reward assignment/modeling, as discussed in Section 1.

*(2) Our segment-level approach improves RLHF training while not suffering from length hacking.* As seen in Table 1, our LM policy performs better than the baselines across all three RLHF policy evaluation benchmarks: AlpacaEval 2.0 (length control win rate LC), Arena-Hard, and MT-Bench. Notably, our model's average response length on AlpacaEval 2.0 and Arena-Hard is not significantly larger than the baseline models', in contrast to the LM policy from the sentence-level approach. Together, these results manifest the merit of our segment-level approach in truly improving the quality of the generated responses while not cheating the benchmark evaluations by response-length hacking (Dubois et al., 2024). As discussed in Section 1, we attribute the gain of our segment-level approach

Figure 3: Examples of dense reward assignment for text sequences encountered in PPO training, comparing our segment-level reward model and the recent token-level design on normal text (**Top**) and text with verbosity/repetition (**Bottom**). Darker color indicates higher reward. In the bottom half, repeated sentences are underlined.

over the baselines to simultaneously achieving both denser reward signals for PPO-based RLHF training and more accurate reward assignment by the design of semantically complete action space.

*(3) Finer action spaces help RLHF training over the classical bandit formulation.* Apart from our denser segment-level approach, in the RLHF policy results in Table 1, we see that the other two finer action space specifications: per-sentence and per-token, both generally improve over the classical design of bandit/sequence-level action space. This provides an extra verification to our motivation of a denser reward signal for RLHF PPO training. Meanwhile, both per-sentence and per-token design can be further refined, respectively by a more break-down action definition and a multi-token action space for more complete semantics of each action, leading to our stronger segment-level design.

*(4) A segment-level reward model is necessary for segment-level reward assignment, vice versa.* One may wonder if we can use the classical bandit reward model to assign segment-level reward in the PPO training. As shown by the results of "Bandit as Segment" in Table 1, this approach performs no better than the classical pure bandit, which in turn under-performs our segment-level design. These comparisons justify the necessity to train a segment-level reward model for segment-level reward assignment. Conversely, using our segment-level reward model to provide only bandit feedback in PPO training ("Segment as Bandit") leads to only marginal gain over pure bandit design. While this further verifies the merit of training and utilizing a segment-level reward model, "Segment as Bandit", however, does not fully benefit from our proposal of a (consistent) segment-level action space. Its weaker results again highlight the gain of denser reward assignment in PPO-based RLHF training.

Appendix B.2 provides generation examples from our main LM policy. Table 6 in Appendix B.1 compares the LM policies in Table 1 on tasks on OpenLLM Leaderboard. Both show that our method, while achieving strong RLHF training, does not suffer from the "alignment tax" (Askell et al., 2021).

### 4.3 ABLATION STUDY

**(a):** *Can our segment-level reward model reasonably distinguish key segments?*

In Fig. 3, we qualitatively compare dense reward assignments from our segment-level reward model and the recent token-level approach on two text sequences appeared in PPO training, representing normal text (*Top*) and broken text with verbosity/repetition (*Bottom*), where repetitions are underlined.

The color blocks in Fig. 3 first verify that our entropy-based approach chops the text into segments with meaningful semantics, while a token can be only part of a word. Further, Fig. 3 confirms that our segment-level reward model assigns higher rewards to key segments in the responses. Meanwhile, we see that the token-level model does not have consistent reward assignment over even a word – often less understandably (only) highly rewards the first letter of a word (*e.g.*, "Siddhartha", "cessation", "tectonic"). The benefit of our desideratum of a semantically complete action space is further testified by the example of repeated sentences in Fig. 3 *Bottom*, where our model assigns a consistent low reward to the repeated sentences, effectively refraining the LM from verbosity/repetition. By contrast,

Table 2: Comparison of different constructions of segment-level reward normalizers. Shown are the results of the resulted PPO-trained LM policies on AlpacaEval 2.0 and Arena-Hard. Best evaluation results are in bold.

| Reward Normalizer | AlpacaEval 2.0 | | | Arena-Hard | |
|---|---|---|---|---|---|
| | LC (%) | WR (%) | # char | WR (%) | # token |
| No Reward Normalization | 7.27 | 2.98 | 448 | 7.1 | 263 |
| Global Statistics of All | 13.88 | 8.32 | 1159 | 12.5 | 411 |
| Statistics of the Last Rewards | 14.55 | 9.69 | 1222 | 15.5 | 459 |
| Regression-based (Section 2.3) | **19.68** | **16.15** | 1622 | **20.0** | 518 |

Table 3: Comparison of different within-segment reward interpolation strategies. Shown are the results of the resulted PPO-trained LM policies on AlpacaEval 2.0 and Arena-Hard. Highest numeric of each metric is in bold.

| Interpolation strategy | AlpacaEval 2.0 | | | Arena-Hard | |
|---|---|---|---|---|---|
| | LC (%) | WR (%) | # char | WR (%) | # token |
| No Interpolation | 15.76 | 8.70 | 1132 | 13.6 | 428 |
| Repeat Segment Reward | 13.64 | 12.80 | **1927** | 15.0 | **546** |
| Even Split (Section 2.3) | **19.68** | **16.15** | 1622 | **20.0** | 518 |

due to the semantic incompleteness of each action, the token-level model still assigns high rewards to several tokens in the repetitions, even in the second repeat, which is undoubtedly undesirable.

**(b):** *How will PPO training perform if we use different constructions of reward normalizers?*

Recall that in our PPO training (Section 2.3), we use simple linear regression to fit location-aware mean and std functions that provide reward normalizers at arbitrary locations of the text sequence. To study if this design is over-engineering, we compare our main method with three simpler constructions of segment-level reward normalizers: *(A)* no reward normalization; *(B)* using the scalar global mean and std over all segment-level rewards in the reward calibration dataset; and *(C)* using the scalar mean and std over the last segment-level rewards in each response of the calibration set, mimicking the normalizers in the classical bandit approach. Table 2 compares the resulted LM policies.

As common RLHF PPO practice, in Table 2, we first see a strong call for reward normalization, without which the training process will be broken. Using global statistics or the statistics of the last segment-level rewards perform similarly, with the latter being slightly better. While policy training under these two normalizer constructions does not break, it is however ineffective, indicating that these two constructions distort the training reward signals to an extent of being (almost) useless. The significantly better performance of our main method over these alternatives verify the necessity of normalizing segment-level rewards by location-aware normalizers able to capture the reward statistics at each arbitrary location (completion portion) of the sampled text sequence, and hence our design of regression-based mean and std functions. Future work may extend these functions with non-linearity.

**(c):** *What will happen if we use different strategies for within-segment reward interpolation?*

Recall from Section 2.3 that, to further densify the learning signal in RLHF for enhancing training, we interpolate the segment-level rewards by evenly splitting the reward of a segment to each of its constituting token. We now compare this even-split interpolation strategy with two other intuitive alternatives: *(A)* no interpolation on the segment-level rewards, use 0 for technical padding in PPO ("No Interpolation"); *(B)* repeat the segment-level reward of a segment to each token in it ("Repeat Segment Reward"). Table 3 shows the performance of the resulted PPO-trained LM policies.

Aligning with our intuition, Table 3 indicates that without any within-segment reward interpolation, the raw segment-level rewards may not be a strong-enough learning signal to incentivize the LM to learn to generate, leading to too-short sequence generations and the subsequent inferior performance. On the other hand, repeating segment-level reward to each of the constituting token results in a too-strong learning signal, where each token has interpolated feedback of the same scale as the feedback to the entire text segment, which has coarser granularity. This undesirable level-up of feedback signal scale, especially the amplification of positive signals on longer segments, may provide a too-strong incentive for the LM policy to learn to generate, making it produce excessively long text sequences. By contrast, the even-split densification strategy in our main method provides interpolated learning signal of a proper scale, which we attribute to the implicit (segment-)length normalization inherited from division by segment length in an even split. Future work may design a proper non-even split.

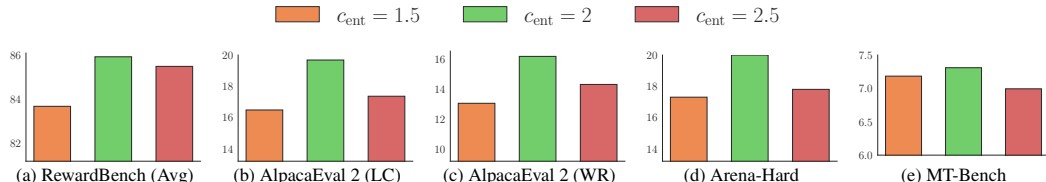

Figure 4: Performance comparison among different entropy cutoffs $c_{\text{ent}}$ for entropy-based text segmentation, comparing the performance of both the resulted reward model and PPO-trained LM policy, both under the same specified $c_{\text{ent}}$. For reward models' performance, we plot the average accuracy over the entire RewardBench.

Table 4: Performance of segment-level reward model on RewardBench when trained by different choices of aggregation function $f(\cdot)$ and different values of temperature $\tau$ in the soft-maximum aggregation Eq. (4). Our main method uses the soft-maximum aggregation function with temperature $\tau = 0.5$.

| Metric | Aggregation Function $f(\cdot)$ | | | Temperature $\tau$ | | |
|---|---|---|---|---|---|---|
| | Soft-maximum | Summation | Average | 0.3 | 0.5 | 0.7 |
| Chat(%) | 97.49 | 97.21 | 96.65 | 97.49 | 97.49 | 97.77 |
| Chat-hard(%) | 55.04 | 55.70 | 52.19 | 56.14 | 55.04 | 55.92 |
| Reasoning(%) | 93.77 | 72.61 | 74.43 | 92.47 | 93.77 | 92.27 |
| Safety(%) | 85.90 | 85.01 | 84.36 | 85.95 | 85.90 | 84.68 |
| Average(%) | 85.93 | 77.96 | 77.32 | 85.53 | 85.93 | 85.29 |

**(d):** *With a different entropy cutoff $c_{\text{ent}}$ for text segmentation, how will our method perform?*

As discussed in Section 4.1, for main results, we use entropy cutoff $c_{\text{ent}} = 2.0$ for entropy-based text segmentation. To investigate the impact of $c_{\text{ent}}$, in Fig. 4, we vary the value of $c_{\text{ent}} \in \{1.5, 2.0, 2.5\}$, and compare the performance of both the resulted reward models and the PPO-trained LM policies.

As seen in Fig. 4, similar to the discussion of token-level approach in Section 1, a smaller $c_{\text{ent}} = 1.5$, which chops text sequence into finer pieces, may result in semantically less complete segments, in turn leading to less accurate reward modelling and the subsequent weaker reward model and LM policy. By chopping text into coarser segments that can be more semantically complete, a larger $c_{\text{ent}} = 2.5$ results in a more accurate reward assignment and thus a higher RewardBench score. Its policy training, however, may be impaired by the sparse reward issue due to prolonged segments, making its LM performance inferior to that from an intermediate $c_{\text{ent}}$ value of 2.0.

**(e):** *What if we use a different aggregation function $f(\cdot)$ in Eq. (4) or another temperature $\tau$?*

Recall that we use the soft-maximum aggregation with temperature $\tau = 0.5$ for constructing the parametrized sequence evaluation Eq. (4) in reward model training. In Table 4 we report the results of reward models trained under two alternative aggregation functions $f(\cdot)$: summation and average; and under different values of $\tau$ in the soft-maximum aggregation.

As demonstrated in Fig. 2 in Section 2.2, chosen responses in our reward training dataset can typically be identified by a few key segments. This translates into the performance gain of soft-maximum aggregation over both summation and average, since the latter two do not highlight the contributions of key segments, but rather focus on average text quality. Meanwhile, we see that our method is relatively robust to the value of $\tau$ in the soft-maximum aggregation, for example, for $\tau \in \{0.3, 0.5, 0.7\}$.

## 5 CONCLUSION AND LIMITATIONS

In this paper, we propose to train and utilize a segment-level reward model for improved RLHF in LMs, motivated by both a denser reward signal in RL-based LM training and semantic completeness of each action for accurate reward assignment. Our method and insight are validated through extensive experiments and ablation studies. Nevertheless, as an early effort in reconsidering the action space in RLHF, our experiments are currently confined to the 3.8B Phi-3 series model, PPO training on a free-form dialog-alike dataset, and instruction-following benchmark evaluations. Our future work includes scaling up to even larger LMs, testing our method on other types of tasks such as math problem solving and code generation, and applying to other RL algorithms, such as REINFORCE.

## ETHICS STATEMENT

On method contributes to the ongoing research on aligning LMs with human preference and values, by proposing a method that aims at improving the effectiveness and efficiency of RLHF in LMs. These improvements can translate to more helpful and less harmful LMs and less compute and resource consumption in model training. Nonetheless, as with previous RLHF works, there is a possibility of misusing our method to train malicious LMs that align with some ill-intended or unethical preference. This potential negative societal impact could be diminished by more closely monitoring the datasets and preference sources where our method is applied.

## REPRODUCIBILITY STATEMENT

To facilitate reproducibility, we elaborate our method in Section 2 and provide a comprehensive algorithm box in Appendix A. We provide details in method implementation and experimental setups in Section 4 and Appendix C. Furthermore, our source code and model checkpoints are anonymously released at https://anonymous.4open.science/r/Segment-Reward-Model-For-PPO-Training-1512 .

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

# Appendix

CONTENTS

## A  ALGORITHM BOX

Algorithm 1 summarizes our method in Section 2 on training the segment-level reward model and utilizing it in PPO-based RLHF LM training. Note that all operations in Algorithm 1 can be efficiently conducted in batch mode, parallel for multiple sample points at once.

---

**Algorithm 1** Training and Utilizing Our Segment-level Reward.

---

**Input:** Binary preference dataset $\mathcal{D}_{\text{pref}} = \{(x, y^w, y^l)\}$ for reward model training, prompt set $\mathcal{D}_{\text{pol}} = \{x\}$ for policy learning, supervised fine-tuned model $\pi_{\text{SFT}}$, reward model training steps $M_{\text{rew}}$, LM policy training steps $M_{\text{pol}}$, entropy cutoff $c_{\text{ent}}$, temperature $\tau$ for the aggregation function, KL coefficient $\beta$ for RLHF PPO training.

**Initialization:** Initialize the segment-level reward model $r_\phi$ and LM policy $\pi_\theta$ from $\pi_{\text{SFT}}$, fix the aggregation function $f(\cdot)$ as the soft-maximum in Eq. (4), initialize other components in the off-the-shelf RLHF PPO routine as specified.

// Training the segment-level reward model
Use $\pi_{\text{SFT}}$ and $c_{\text{ent}}$ to split the responses $\{(y^w, y^l)\}$ in $\mathcal{D}_{\text{pref}} = \{(x, y^w, y^l)\}$ into segments.
**for** iter $\in \{1, \ldots, M_{\text{rew}}\}$ **do**
    Sample a minibatch $\mathcal{B} = \{(x_i, y_i^w, y_i^l)\}_i \sim \mathcal{D}_{\text{pref}}$.
    With $f(\cdot)$ and $\tau$, calculate $e_\phi(x_i, y_i^w)$ and $e_\phi(x_i, y_i^l)$ by Eq. (4) for $(x_i, y_i^w, y_i^l) \in \mathcal{B}$.
    Optimize reward model $r_\phi$ by Eq. (3).
**end for**

// Utilizing the segment-level reward model in PPO-based LM policy learning
Estimate the reward normalizer functions $\text{Mean}(p)$ and $\text{Std}(p)$ as described in Section 2.3.
**for** iter $\in \{1, \ldots, M_{\text{pol}}\}$ **do**
    Sample a minibatch $\mathcal{B} = \{x_i\}_i \sim \mathcal{D}_{\text{pol}}$.
    Sample a response $y_i \sim \pi_\theta(\cdot \mid x_i)$ for each $x_i \in \mathcal{B}$
    Use $\pi_{\text{SFT}}$ and $c_{\text{ent}}$ to segment each $y_i$; record the completion portion $p$ of each segment.
    Use $r_\phi$ to assign a segment-level reward to each segment $a_t$ in each $y_i$
    Normalize each segment reward $r_\phi(s_t, a_t)$ as $r_\phi(s_t, a_t) \leftarrow (r_\phi(s_t, a_t) - \text{Mean}(p))/\text{Std}(p)$.
    Interpolate $r_\phi(s_t, a_t)$ to each token $y_i$, as $\forall a_t \in y, \forall y_i \in a_t, \tilde{r}_\phi([x, y_{<i}], y_i) = r_\phi(s_t, a_t)/|a_t|$
    With KL coefficient $\beta$, optimize policy LM $\pi_\theta$ against $\tilde{r}_\phi$ by the PPO routine.
**end for**

---

## B  ADDITIONAL RESULTS

### B.1  TABULAR RESULTS

Table 5: Break down classification accuracy on RewardBench to test reward model's quality, comparing the reward models in our main result table Table 1 in Section 4.2.

| Action Definition | Average(%) | Chat(%) | Chat-hard(%) | Reasoning(%) | Safety(%) |
|---|---|---|---|---|---|
| Bandit | 81.11 | 97.77 | 57.68 | 77.54 | 86.00 |
| Sentence | 84.42 | 97.49 | 55.48 | 89.35 | 85.87 |
| Token | 81.21 | 98.04 | 54.39 | 79.95 | 84.03 |
| Segment | 85.93 | 97.49 | 55.04 | 93.77 | 85.90 |

Table 6: Evaluation results of downstream tasks on the HuggingFace OpenLLM Leaderboard (Beeching et al., 2023), comparing LM policies in Table 1.

| Action Definition | ARC | TruthfulQA | Winograd | HellaSwag | MMLU | GSM8K | Average |
|---|---|---|---|---|---|---|---|
| SFT (No RLHF) | 63.65 | 51.89 | 75.06 | 77.99 | 70.34 | 79.50 | 69.74 |
| Bandit (Sequence) | 63.82 | 52.25 | 74.51 | 78.13 | 70.37 | 78.90 | 69.66 |
| Sentence | 63.90 | 52.41 | 73.48 | 78.20 | 70.34 | 80.10 | 69.74 |
| Token | 63.22 | 51.71 | 73.56 | 78.43 | 70.28 | 80.40 | 69.60 |
| Segment (**Ours**) | 63.82 | 52.42 | 74.19 | 78.27 | 70.32 | 79.00 | 69.67 |
| Bandit as Segment | 63.14 | 51.06 | 72.38 | 78.20 | 70.41 | 80.40 | 69.27 |
| Segment as Bandit | 63.99 | 52.47 | 73.72 | 78.12 | 70.51 | 79.90 | 69.79 |

## B.2 EXAMPLES OF GENERATED RESPONSES

Below, we provide examples of generated responses from our LM policy in the main results (Table 1) on the benchmarks AlpacaEval 2.0 (first two examples) and Arena-Hard (last two examples).

---

**Prompt:** When are hops added to the brewing process?

**Response:** Hops are typically added to the brewing process during the boil, which is a critical phase in beer production. The boiling process plays several important roles in beer making, including the extraction of hop bitterness, flavor, and aroma compounds, as well as the sterilization of the wort (the liquid extracted from the mashing process that contains the sugars to be fermented). Here is a breakdown of when hops are generally added during the boil:

1. **Early Boil (First 30 minutes)**: Hops can be added early in the boil, and this addition typically contributes more to the bitterness of the beer. The longer the hops are boiled, the more bitterness they impart, due to the isomerization of alpha acids. Early additions might be used to start the boiling process and can also contribute to some of the beer's flavor and aroma.

2. **Middle Boil (Later 30 minutes)**: Hops added during the middle of the boil contribute more to the beer's flavor than bitterness. This timing allows for the extraction of essential oils and resins that provide a range of hop-derived flavors and aromas without overly increasing the bitterness.

3. **Late Boil (Final 10-15 minutes)**: Hops added in the last 10-15 minutes of the boil contribute the most to the beer's aroma. The shorter boil time for these hops reduces the extraction of alpha acids, minimizing bitterness while maximizing the retention of essential oils and volatile aroma compounds. This is the stage where many brewers add hops to achieve hoppy aroma profiles in styles like IPAs, double IPAs, and American pale ales.

4. **Dry Hopping (After Fermentation)**: While not part of the traditional boiling process, dry hopping is a modern technique used to add aroma and flavor without significantly increasing the beer's bitterness. Hops are added to the fermenter after primary fermentation has ceased. This process enhances the beer's aromatic qualities by infusing it with the intense, floral, fruity, or spicy notes that hop varieties offer.

The timing of hops additions is crucial and varies based on the beer style, the desired bitterness level, and the aromatic profile the brewer aims to achieve. Experimentation and experience play significant roles in determining the optimal hop schedule for a particular brew.

---

**Prompt:** Introduce the "financial markets and institutions" by Frederic S. Mishkin.

**Response:** Frederic S. Mishkin is a prominent economist with significant contributions to the field of financial markets and institutions. In his influential book, "The Economics of Money, Banking, and Financial Markets," Mishkin delves into the intricate workings of financial markets and the roles of financial institutions in the economy.

Financial markets encompass the platforms where buyers and sellers of financial assets, such as stocks, bonds, and derivatives, interact to determine prices and execute trades. These markets play a crucial role in efficiently allocating resources, facilitating risk management, and providing a platform for price discovery.

Financial institutions, on the other hand, are intermediaries that facilitate the flow of funds between borrowers and lenders. They encompass a wide range of entities, including banks, investment firms, insurance companies, and pension funds. These institutions play a vital role in the functioning of financial markets by providing various services, such as credit provision, investment advice, and risk management.

Mishkin emphasizes the importance of a well-functioning financial system in promoting economic growth and stability. He highlights the key roles of financial markets and institutions in promoting savings, investments, and efficient allocation of capital. Furthermore, he explores the mechanisms through which financial markets and institutions contribute to economic stability by facilitating liquidity and risk management.

In his work, Mishkin also discusses the potential risks and vulnerabilities in the financial system, such as asset price bubbles, excessive leverage, and systemic risk. He acknowledges the role of financial markets and institutions in amplifying these risks and emphasizes the importance of prudent regulation and oversight to safeguard the stability of the financial system.

Overall, Frederic S. Mishkin's introduction to financial markets and institutions provides a comprehensive understanding of the complex dynamics and crucial functions of these entities in the modern economy. His work serves as an essential resource for economists, policymakers, and students seeking to grasp the intricacies of financial markets and institutions.

**Prompt:** Provide me with a breakfast recipe that is quick to make and is high in protein (at least 30 grams) and has a variety of ingredients.

**Response:** Here is a quick and high-protein breakfast recipe that includes a variety of ingredients:

**High-Protein Breakfast Quinoa Bowl**

Ingredients:

- 1/2 cup quinoa

- 1 cup water

- 1/2 cup Greek yogurt (plain, full-fat)

- 1/4 cup chopped nuts (almonds, walnuts, or pecans)

- 1/4 cup fresh berries (strawberries, blueberries, or raspberries)

- 1 tablespoon chia seeds

- 1 tablespoon honey or maple syrup (optional)

- A pinch of salt

1. Rinse the quinoa under cold water to remove its natural coating, saponin, which can make it taste bitter or soapy.

2. In a small saucepan, bring the water to a boil. Add the quinoa and a pinch of salt. Reduce heat to low, cover, and simmer for about 15 minutes, or until the water is absorbed and the quinoa is fluffy.

3. While the quinoa is cooking, prepare the toppings. Warm the Greek yogurt in the microwave for about 30 seconds or until it's at a comfortable temperature.

4. In a bowl, mix the cooked quinoa with the Greek yogurt, ensuring the quinoa is well coated.

5. Add the chopped nuts and fresh berries on top of the quinoa-yogurt mixture.

6. Sprinkle chia seeds over the bowl for an extra protein boost and a dose of omega-3 fatty acids.

7. Drizzle honey or maple syrup over the bowl for a touch of sweetness, if desired.

8. Serve immediately and enjoy your high-protein, nutrient-rich breakfast!

This breakfast bowl provides a good balance of carbohydrates, protein, and healthy fats, making it a satisfying and nutritious start to the day. The quinoa is a complete protein, meaning it contains all nine essential amino acids, while the Greek yogurt adds additional protein and probiotics for gut health. The nuts and berries offer healthy fats, antioxidants, and fiber.

**Prompt:** Write a python function that solves a quadratic equation.

**Response:** To solve a quadratic equation of the form $ax^2 + bx + c = 0$, we can use the quadratic formula:

$$x = \frac{-b \pm \sqrt{b^2 - 4ac}}{2a}$$

Here's a Python function that implements this formula:

```python
import math

def solve_quadratic(a, b, c):
    """
    Solves a quadratic equation of the form ax^2 + bx + c = 0.

    Parameters:
    a (float): Coefficient of x^2
    b (float): Coefficient of x
    c (float): Constant term

    Returns:
    tuple: A tuple containing the two solutions (x1, x2)
    """
    # Calculate the discriminant
    discriminant = b**2 - 4*a*c

    # Check if the discriminant is positive, negative, or zero
    if discriminant > 0:
        # Two real and distinct roots
        x1 = (-b + math.sqrt(discriminant)) / (2*a)
        x2 = (-b - math.sqrt(discriminant)) / (2*a)
        return (x1, x2)
    elif discriminant == 0:
        # One real root (repeated)
        x = -b / (2*a)
        return (x,)
    else:
        # No real roots (complex roots)
        real_part = -b / (2*a)
        imaginary_part = math.sqrt(-discriminant) / (2*a)
        return (complex(real_part, imaginary_part), complex(real_part, -
            imaginary_part))

# Example usage
a, b, c = 1, 5, 6
solutions = solve_quadratic(a, b, c)
print(f"The solutions are: {solutions}")
```

This function first calculates the discriminant $(b^2 - 4ac)$. Based on the value of the discriminant, it determines the nature of the roots (real and distinct, real and repeated, or complex) and calculates the roots accordingly. For complex roots, it returns a tuple of complex numbers.

Remember, the coefficients $a$, $b$, and $c$ should be provided as floating-point numbers to handle cases where they might not be integers.

## C MORE IMPLEMENTATION DETAILS

Table 7: Model hyperparameters used in reward model training.

| Hyperparameter | Value |
|---|---|
| Batch Size | 128 |
| Training Epochs | 1 |
| Max prompt length | 1728 |
| Max total length | 2048 |
| DeepSpeed ZeRO stage | 3 |
| Optimizer | Adam |
| Learning Rate | 1e-6 |
| Gradient clipping norm | 1.0 |
| Entropy threshold $c_{\text{ent}}$ | 2.0 |
| Soft-maximum temperature $\tau$ | 0.5 |

Table 8: Model hyperparameters used in PPO-based LM policy training.

| Hyperparameter | Value |
|---|---|
| Batch Size | 128 |
| Rollout batch size | 1024 |
| Micro rollout batch size | 16 |
| Training Epochs | 1 |
| Max prompt length | 1024 |
| Max generation length | 1024 |
| DeepSpeed ZeRO stage | 2 |
| Actor learning rate | 5e-7 |
| Critic learning rate | 9e-6 |
| Gradient clipping norm | 1.0 |
| Entropy threshold $c_{\text{ent}}$ | 2.0 |
| Soft-maximum temperature $\tau$ | 0.5 |
| Value clipping | 0.25 |
| KL coefficient $\beta$ | 0.01 |

**Implementation Details.** We tabulate detailed parameter settings in Table 7 and Table 8. Most of them are the same as the default setting in OpenRLHF. Both the reward model and PPO training employ the Adam optimizer (Kingma & Ba, 2014), with $\beta_1 = 0.9$ and $\beta_2 = 0.95$. To save GPU memory, we use gradient checkpointing (Chen et al., 2016) and flash attention (Dao et al., 2022).

For reward model training, we set the maximum prompt sequence length as 1792 tokens, with the total sequence length (including both prompt and response) capped at 2048 tokens. During data preprocessing, we apply left truncation to the prompt and right truncation to the response. If the EOS token in the response is truncated, we manually change the last token in the truncated response to the EOS token. The global mini batch size for reward model training is set to 128, with each GPU processing a micro batch size of 8. To facilitate distributed training, we utilize DeepSpeed ZeRO-3. For our segment-level reward model, we set the entropy threshold $c_{\text{ent}} = 2.0$. The baseline bandit reward model is technically implemented as setting the entropy threshold $c_{\text{ent}} = 1000$, restricting reward computation to the EOS token only, while the baseline token-level reward model is implemented as setting the entropy threshold $c_{\text{ent}} = 0$, ensuring that a reward is computed for each token in the text sequence.

For PPO training, we set the replay buffer size (`rollout_batch_size`) to 1024 and the batch size per GPU for generation (`micro_rollout_batch_size`) to 16. The maximum prompt sequence length is set as 1024 tokens, and the maximum generated sequence length is also set to 1024 tokens. In PPO's on-policy sampling, for each prompt in the mini-batch, a single response is sampled via top-$p$ sampling with $p = 1.0$ and sampling temperature 1.0. We use DeepSpeed ZeRO-2 for distributed training. The actor learning rate is set to the default value of $5 \times 10^{-7}$, and the critic learning rate is also the default value of $9 \times 10^{-6}$. The clipping coefficient for value loss (value clip) is set to 0.25 for PPO training based on segment- and token-level reward model , and as default to 0.2 for bandit-reward-based PPO training. The clipping coefficient for policy loss (eps clip) is set to 0.2. The KL coefficient is kept to the default value of $\beta = 0.01$.

**Chat Template.**  Below is the chat template used for PPO training, reward model training, and evaluation sampling with Phi-3. The original Phi-3-Mini chat template includes an additional EOS token, which we removed in experiments due to incompatibility, leaving only the <|end|> token.

---

<|user|>

{the user query} <|end|>

<|assistant|>

{Assistant Response} <|end|>

---

## D  MORE RELATED WORK

**Learning-from-preference.**  Learning-from-preference classically takes a two-stage approach where a reward model is first trained on a dataset of binary or multiple ranking via maximizing the choice model likelihood (Bradley & Terry, 1952; Plackett, 1975; Luce, 2012), before optimizing the RL/control policy against the learned reward model by RL algorithms (Akrour et al., 2011; 2012; Fürnkranz et al., 2012). Earlier application in deep learning mainly focuses on relatively simple neural-network policy for robotics/control tasks (Christiano et al., 2017; Ibarz et al., 2018; Bıyık et al., 2019; Brown et al., 2019; 2020; Lee et al., 2021; Shin et al., 2021; Hejna & Sadigh, 2023a;b). Implanting its success in robotics, in natural language generation, this two-stage learning-from-preference paradigm has been scaled up and popularized in the post-training stage to align LMs with specific human values, with applications ranging from text summarization (Ziegler et al., 2019; Stiennon et al., 2020), prompt generation (Yang et al., 2023), to (task-oriented) conversational agent (*e.g.*, Ouyang et al., 2022; Bai et al., 2022a; Menick et al., 2022; Feng et al., 2023; OpenAI, 2023).

To alleviate the complexity in fitting an explicit reward model, motivated by the theory of maximum-entropy control and RL (Ziebart et al., 2008; Ziebart, 2010; Finn et al., 2016), direct preference optimization methods (DPO, *e.g.*, Rafailov et al., 2023; Tunstall et al., 2023; Azar et al., 2023; Yuan et al., 2023; Zhao et al., 2023; Ethayarajh et al., 2024; Yin et al., 2024) were recently proposed to directly train LMs on a preference dataset by using their log-density-ratio as the classification logit, which have been adapted to train text-to-image diffusion models (*e.g.*, Wallace et al., 2023; Yang et al., 2024; Li et al., 2024b; Gu et al., 2024).

In this paper, we contribute to the literature of learning-from-preference by re-thinking a suitable definition of action space in the RL formulation of LM generation and preference alignment. Motivated by semantic completeness in linguistics, we define each action as "text segment", spanning across a small amount of tokens and interpolating between prior works' action space of either the finest "per token" or the coarsest "full sequence". With this design, our method may benefit from both denser reward signal for easier RL-based LM training and the semantic completeness of each action for more accurate training guidance, as experimentally verified in Section 4.

**Training Signals for RL-based Language Model (LM) Training.**  In RL-based LM fine-tuning, a classical training signal for adapting LMs to the specific downstream task is the native trajectory-level downstream test metrics (*e.g.*, Ryang & Abekawa, 2012; Ranzato et al., 2015; Rennie et al., 2017; Paulus et al., 2017; Shu et al., 2021; Lu et al., 2022). This approach intrinsically uses a bandit formulation of LM generation that treats the entire generated sequence as a single action. As discussed in Section 1, ignoring the sequential nature of LM generation, this bandit training signal delays the feedback to each token/phrase selection, and can thus incur optimization difficulty (Guo et al., 2022; Snell et al., 2022). With various forms of stronger data or compute requirements, task-specific per-step training signals have been proposed to mitigate this sparse reward issue. Assuming abundant golden expert data for supervised (pre-)training, Shi et al. (2018) construct per-step reward via inverse RL (Russell, 1998); Guo et al. (2018) use a hierarchical approach; Yang et al. (2018) learn LM discriminators; Lin et al. (2017) and Yu et al. (2017) use the expensive and high-variance Monte Carlo rollout to estimate per-step reward from a sequence-level adversarial reward function trained in the first place; while Le et al. (2022) use some rule-based intermediate training signal derived from the oracle sequence-level evaluation, without explicitly learning per-step reward.

Similarly, in RLHF, to move forward from the classical bandit formulation, methods have recently been proposed to ground sparse preference labels into dense per-step feedback, with applications

in task-oriented dialog systems (*e.g.*, Ramachandran et al., 2021; Feng et al., 2023) and variable-length text-sequence generation (Yang et al., 2023). Our paper seeks to reconcile dense *v.s.* sparse training signal in RLHF by distributing feedback to the level of semantically complete "text segment", interpolating between the densest "token level" and the sparsest "sequence level" and ideally getting the benefit of both worlds: easier RL training and accurate optimization signal. Meanwhile, as seen in Section 2, our method adheres to the classical two-stage RLHF paradigm without requiring extra data or compute.

**Other LM Preference Alignment Methods.** Apart from RL methods, strategies have been developed to align LMs with preference by adding external filters on top of the pretrained LMs, for safety checking the generations or the training texts (*e.g.*, Xu et al., 2020). Vanilla maximum likelihood estimation has also been adopted for training LMs on curated datasets (Hancock et al., 2019; Solaiman & Dennison, 2021; Scheurer et al., 2022), or instruction fine-tuning LMs on massive highly-curated sets of tasks (Sanh et al., 2022; Chung et al., 2022). With extra requirements on data, modelling, and/or compute, recent LM works also conduct preference alignment by formulating text generation as a constraint satisfaction problem on LM's generation distribution (*e.g.*, Khalifa et al., 2021; Korbak et al., 2022; Go et al., 2023), or utilizing the preference dataset in LMs' pre-training stage (Korbak et al., 2023) .

In this paper, we seek to refine RL-based LM preference alignment by re-thinking the suitable action space in the RL formulation that allows both denser immediate feedback while not jeopardizing the feedback accuracy. Our segment-level design is validated through numeric and example in Section 4.

# E   MORE ON THE REWARD NORMALIZERS IN PPO TRAINING

To center the assigned rewards from the reward model and reduce their variance, in most open-sourced (bandit) RLHF PPO implementation (*e.g.*, Havrilla et al., 2023; Hu et al., 2024), the bandit reward of the newly sampled response $y$ is first "Z-score" normalized, before being fed into the PPO routine. Concretely, for the prompt $x$ and sampled response $y$, the bandit reward $r_\phi(x, y)$ is normalized as $r_\phi(x, y) \leftarrow (r_\phi(x, y) - \mu)/\sigma$, where $\mu$ and $\sigma$ are respectively the mean and standard deviation of (bandit) rewards in the reward calibration dataset. The PPO routine starts by using this normalized $r_\phi(x, y)$, *e.g.*, first subtract it by the KL regularization, and then calculate the advantage estimates and value function training target, *etc*.

For the segment-level action space, we will then need to normalize the reward $r_\phi(s_t, a_t)$ for each segment $a_t$. As shown in Table 2 ("Global Statistics of All"), the most intuitive idea of simply using the global mean and standard deviation over all segment-level rewards in the reward calibration dataset does not train a good LM. Looking into the responses sampled in PPO training and in the reward calibration dataset, we find that, for example, the beginning segments of the responses are typically greeting alike phases that are less informational and/or essential to respond to the given prompt, and hence have relatively lower rewards. If we normalize the segment-level rewards of those early segments by the global mean and standard deviation, those normalized rewards will be significantly negative, rather than centered around 0. This will undesirably refrain the generation of necessary greeting alike phases, resulting in an "impolite LM" and thus inferior benchmark results. More generally, the linguistic structure of the responses leads to certain correlation between the mean and standard deviation of segment-level reward values and the location of segment in the response, *e.g.*, in the early/middle/later part. This observation motivates us to design location-aware reward normalizers that can approximately capture the reward statistics at an arbitrary location of the response, so that the normalized segment-level rewards can be more centered and less varying. It is important to have proper reward normalizers at an *arbitrary* location of the response, because the response sampled in PPO training will have a stochastic total length, nondeterministic number of segments, and less-controllable length of each segment. These considerations motivate our design of the regression-based reward normalizer functions in Section 2.3.

# F   MORE ON THE EVEN-SPLIT REWARD INTERPOLATION STRATEGY

In this section, we expand our previous discussion in Section 4.3 **(c)**, on **(1)** a plausible reason why *no reward interpolation* does not work well; and **(2)** the even-split reward interpolation strategy in Section 2.3 *does not conflict* with our segment-level design for LM's RLHF. For notation simplicity, all segment-level rewards $r_\phi(s_t, a_t)$ in this section are after normalization.

As discussed in Section 4.3 **(c)**, for the variant of without reward interpolation, we follow the classical bandit RLHF implementation to pad 0 for the "reward" of intermediate token within a segment. Similar to the bandit implementation, the array of KL-regularized RL training-signal under our segment-level rewards takes the form of, *for example*,

$$[-\mathrm{KL}_1, -\mathrm{KL}_2, r_\phi(s_1, a_1) - \mathrm{KL}_3, -\mathrm{KL}_4, r_\phi(s_2, a_2) - \mathrm{KL}_5, \ldots],$$

where $\mathrm{KL}_i$ denotes the token-wise factorization of the KL regularization term in the PPO objective. Since $\mathrm{KL}(\cdot||\cdot) > 0$, KL-regularized RL training-signals are negative for those intermediate tokens within a segment. Note that, by design, there are more those intermediate tokens than tokens with a segment-level reward $r_\phi(s_t, a_t)$. These factors, especially a lot of negative training signals in generation, can refrain the LM from learn to generate, as seen by the significantly shorter generation lengths that the "No Interpolation" variant shows in Table 3.

By contrast, with our even-split reward interpolation strategy, the array of KL-regularized RL training-signal under our segment-level rewards now takes the form of, again *for example*,

$$\left[\frac{r_\phi(s_1, a_1)}{3} - \mathrm{KL}_1, \frac{r_\phi(s_1, a_1)}{3} - \mathrm{KL}_2, \frac{r_\phi(s_1, a_1)}{3} - \mathrm{KL}_3, \frac{r_\phi(s_2, a_2)}{2} - \mathrm{KL}_4, \frac{r_\phi(s_2, a_2)}{2} - \mathrm{KL}_5, \ldots\right].$$

The training signals to intermediate tokens are now $r_\phi(s_t, a_t)/|a_t| - \mathrm{KL}_i$, which will be positive/less-negative at least for good segments in the responses (the segments $a_t$ with high reward $r_\phi(s_t, a_t)$), due to reward normalization. These positive/less-negative training signals could incentivize the LM to learn to properly generate, as seen by the adequate generation lengths that our main "Even Split" variant presents in Table 3.

We note that our even-split reward interpolation strategy *does not* conflict with our segment-level design: *every token within the same segment receives the same "evenly-split reward"* $r_\phi(s_t, a_t)/|a_t|$. By contrast, in the token-level design, every token $y_i$ will, in general, have a different reward $r([x, y_{<i}], y_i)$. See Fig. 3 for a pictorial illustration of such a difference in reward assignment. In Fig. 3, it is clear that the token-level reward assignment can be overly granular — in many cases are inconsistent even within a word, which is counter-intuitive and less desirable.

To sum up, by the semantic completeness of each action, our design of segment-level MDP facilitates more accurate and consistent reward assignments, compared to the token-level MDP. This benefit will *not* be broken by our even-split reward interpolation strategy. By the previous discussion on KL-regularized RL training-signal, we regard this strategy as a useful technique to cope with the per-token KL regularization in RLHF PPO training, which is extraneous to RLHF reward modeling/assignment.

## G PPO TRAINING CONVERGED IN ONE EPOCH

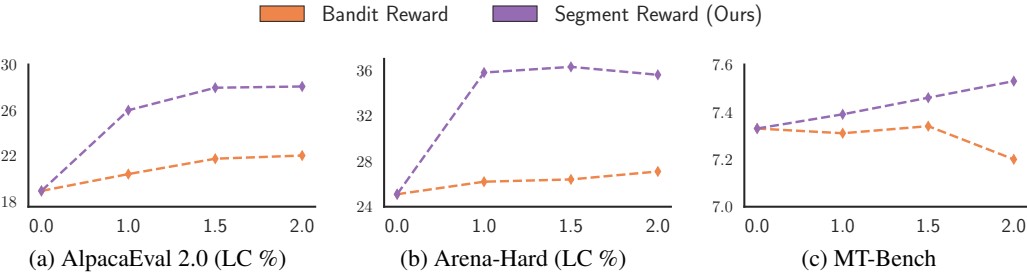

(a) AlpacaEval 2.0 (LC %)    (b) Arena-Hard (LC %)    (c) MT-Bench

Figure 5: Training curves of the (PPO-trained) LM policies on the three tested RLHF benchmarks, comparing training under our segment-level reward model and the classical bandit reward model. Both policies are trained for two epochs — one epoch more than our main results. $x$-axis represents the number of training epochs, *e.g.*, 1.5 represents the performance of the checkpoints after training for 1.5 epochs. $y$-axis is the benchmark score.

For our main experiment results (Section 4.2), we follow the default setting in OpenRLHF to train all models by PPO for one epoch. To verify that the training converged, in Fig. 5, we train the LM policies under our segment-level reward and the classical bandit reward for two epochs, and evaluate the intermediate policy checkpoints on the three tested RLHF benchmarks. The backbone model here is Phi3-mini-4k-instruct.

In accordance with the default in OpenRLHF, in Fig. 5, we see that the training under our segment-level reward and the classical bandit reward (approximately) converged in one epoch — further training may in fact deteriorates certain policy performance. In Fig. 5, it is clear that policy training under our segment-level reward is consistently better than the classical bandit reward, over the entire training process. Fig. 5 also helps to justify the performance comparison in our main results (Table 1).

