# OpenReview forum: "Segmenting Text and Learning Their Rewards for Improved RLHF in Language Models"
_ICLR.cc/2025/Conference — Submitted to ICLR 2025_

### Official Review · Reviewer_JzqS · 2024-10-24

**Soundness:** 1
**Presentation:** 2
**Contribution:** 2
**Rating:** 3
**Confidence:** 4

**Summary:**

The paper proposes a new approach to improve Reinforcement Learning from Human Feedback (RLHF) for language models by using a segment-level reward model. This method strikes a balance between dense token-level rewards and sequence-level bandit rewards. Rather than assigning rewards to individual tokens (which may lack semantic meaning) or entire sequences (which leads to sparse rewards), the model assigns rewards to semantically meaningful text segments. The authors also introduce techniques for dynamic text segmentation, segment-based reward learning, and novel reward normalizers that adapt to different points in a sequence. The approach shows competitive performance on RLHF benchmarks and demonstrates improved sample efficiency and reward assignment compared to previous methods.

**Strengths:**

(1) The use of a segment-level reward model addresses the limitations of both bandit and token-level approaches, allowing for more meaningful and semantically complete feedback, which can enhance the training signal.

(2) By implementing dynamic segmentation based on the entropy of the model's predictive distributions, the method can effectively identify semantically coherent segments, improving the accuracy of reward assignments.

(3) The approach mitigates the sparse reward issue commonly found in RLHF by providing denser feedback through segment-level rewards, potentially leading to more efficient learning and better policy optimization.

**Weaknesses:**

(1) The results are not convincing. The author should utilize larger models like llama3-8B to further illustrate their contributions.
(2)  why not conduct experiments on RLHF models like phi-dpo?
(3) Segment-level does not intuitive to me. I consider larger models have solved the challenged the author purpose.

**Questions:**

See the weakness section.

---

> ### Author Response · Authors · 2024-11-22
> **Response to Reviewer JzqS**
>
> We thank Reviewer **JzqS** for constructive feedback.
> Please would you check our general response for our additional experiment results and discussions.
> Below are our detailed responses to your concerns.
>
>
> > **Q1.**
> Use Llama3-8B and “RLHF’ed” model as the backbone model.
>
> **A.**
> Thank you so much for this great suggestion!
>
> In General Response **Q1** and **Q2** (https://openreview.net/forum?id=cK7yrw5g5Q&noteId=2hOU6TzlOB), we present the experiment results per your suggestion. It is clear that our segment-level design again outperforms both the classical bandit approach and the recent token-level design, when using Llama3-8B-SFT or the “RLHF’ed” Phi3-mini-4k-instruct as the backbone model.
>
>
> > **Q2.**
> Larger models have solved the challenges that the authors purpose.
>
> **A.**
> We respectfully request the reviewer to clarify this concern and provide more information pertaining to which and why larger models have solved the challenges that we identified or purposed.
>
> Without further information, we respectfully disagree with the reviewer and believe that the issue of an oversubtle action space in token-level design and the sparse reward issue in sequence-level/bandit approach are both orthogonal to model size. As can be seen in our additional experiment results in General Response **Q1** and **Q3**, the advantages of our segment-level design over bandit and token-level approaches extend to the larger 8 billion parameter Llama3-8B-SFT backbone. In General Response **Q2**, we also see that our method can more significantly improve the already “RLHF’ed” Phi3-mini-4k-instruct model, compared to the bandit and token-level designs.
>
> The increasing model sizes and exhausting training corpus call for a more effective utilization of the data. As such, we believe that our method can be a strong ingredient to the post-training pipeline to more effectively train stronger larger models. And our method’s strong efficacy may be particularly helpful when training data are limited, such as RLHF on a specific subdomain.

---

> > ### Comment · Reviewer_JzqS · 2024-11-22
> >
> > Thank you for your rebuttal. However, the results are still not convincing to me. I do not see any practical value in the proposed method even though the authors claim that they achieve a bit of improvement compared with vanilla models. When the model is trained using ultrafeedback dataset (DPO/PPO/SimPO), the performance would be significantly better than your proposed method, whether it's for alpacaeval or arena-hard. So I do not see the actual value in practical downstream post-training. If the authors can address that the methods can adapt to those post-alignment models and achieve substantial improvement, I may reconsider the score currently I make. Currently, I tend to keep the score.

---

> > > ### Author Response · Authors · 2024-11-22
> > > **Further Response to Reviewer JzqS**
> > >
> > > Thank you so much for your quick reply! We respectfully hypothesize that you may misunderstand our main experiment results (Table 1 in the paper) and the additional experiment in the General Response (https://openreview.net/forum?id=cK7yrw5g5Q&noteId=2hOU6TzlOB).
> > >
> > > As discussed in L57-60, in this paper, we seek to improve the RLHF PPO training in LMs by re-considering the granularity of RL action space underlying the PPO training. Specifically, we propose the action space of (semantically complete) text segments, which can be denser than the classical bandit/sequence-level action space for RLHF PPO training while avoiding being oversubtle as in the recent token-level MDP in RLHF PPO (demonstrated in Figure 3).
> > >
> > > Therefore, our method proposed in this paper can be used as a direct substitution to the classical (bandit action space) PPO routine in the post-training pipeline [e.g., 1]. In our main and additional experiments, we have applied our method onto the PPO training for RLHF post-alignment and compared with baselines thereof, with training prompts from the ultrafeedback dataset (L296-298).
> > >
> > > As discussed in Section 4.2 and General Response, our main baselines are the LM policy trained by RLHF PPO under the classical bandit action space (referred to as “Bandit (Sequence)”) and the LM policy trained by RLHF PPO under the recent token-level action space (referred to as “Token”). We did not claim that the vanilla raw model without PPO training is our main baseline.
> > >
> > > In Table 1 of the paper and Tables A & C in the General Response, it is clear that the RLHF PPO training under our segment-level action space results in significantly stronger LM policies compared to the LM policies trained by RLHF PPO under the bandit or token-level action space, in all the three popular RLHF benchmarks: AlpacaEval 2.0 (length-controlled win rate), Arena-Hard (win rate) and MT-Bench. Further, in Table B of the General Response, we further show that the RLHF PPO training under our segment-level design can again be significantly more performant than the RLHF PPO under the bandit or token-level action space, in the practical setting where the backbone model has already been SFT and DPO tuned.
> > >
> > > Based on the above clarification, we would like to argue that our method can be a strong improvement to the existing RLHF PPO training routines, especially the ones based on the classical bandit or the recent token-level action space. We believe that our experiment results can demonstrate the significant practical value of our proposed method.
> > >
> > >
> > > [1] A popular implementation of bandit RLHF PPO in LMs is OpenRLHF: https://github.com/OpenRLHF/OpenRLHF

---

> ### Author Response · Authors · 2024-11-25
> **Follow up with Reviewer JzqS**
>
> Dear Reviewer JzqS,
>
> We are deeply grateful for your insightful suggestions!
>
> As the rebuttal period will end soon, we would like to double check if our responses can address your concerns on the effectiveness of our method on Llama3-8B and already “RLHF’ed” models, the practical value of our method in LLMs’ post-training pipeline, or any additional concerns.
> Please kindly let us know if you have any remaining questions so that we can address them during the remaining discussion period.
>
> If our responses have addressed your concerns, please could you re-evaluate our work based on the updated information?

---

> ### Author Response · Authors · 2024-12-01
> **We are looking forward to further discussions!**
>
> Dear Reviewer JzqS,
>
> We sincerely appreciate the time and effort you dedicated to reviewing our paper, and the invaluable suggestions that you raised.
>
> For your main concern on the practical value of our method in the post-training/RLHF pipeline, we believe that our main response (https://openreview.net/forum?id=cK7yrw5g5Q&noteId=jxeKXxAB2D), further response (https://openreview.net/forum?id=cK7yrw5g5Q&noteId=9izSittFJ1), and additional experiments (https://openreview.net/forum?id=cK7yrw5g5Q&noteId=2hOU6TzlOB) can effectively clarify your concern.
>
> As a quick summary:
> - The goal of our paper is to improve RLHF PPO training by re-considering the granularity of RL action space for LMs.
> - In all experiments, we directly compared with the standard RLHF PPO routine (i.e., the bandit formulation) and the recently proposed token-level formulation of RLHF PPO.
> - Our gain over these two PPO routines indicates that our method can be directly used as a substitution of the standard PPO routine in the post-training pipeline.
> - Furthermore, our strong experiment results on the already “RLHF’ed” backbone model demonstrate our method’s ability of further improving the already DPO’ed/instruction-tuned model.
>
> We understand that you have a busy schedule and are deeply grateful for your suggestions. As the (extended) Author-Reviewer discussion period will end tomorrow, we are eager to know if our responses can adequately address your concerns and if there are any additional questions that you’d like to discuss. We will greatly appreciate the opportunity to hear from you and further discuss our paper if needed
>
> Thank you again for your constructive feedback!
>
> All the best,
>
> Authors

---

### Official Review · Reviewer_aY55 · 2024-10-31

**Soundness:** 3
**Presentation:** 3
**Contribution:** 2
**Rating:** 6
**Confidence:** 4

**Summary:**

The paper formulates RLHF on segment-level MDP. The authors propose a method to train a segment-level reward model, and correspondingly use it on online RL methods, i.e. PPO. They also improve the original PPO pipeline with two modifications: 1) a new reward normalization method; 2) a way to assign token-level reward from segment-level reward. The experiments, especially ablation studies, are comprehensive to validate their implementation choices.

**Strengths:**

1. The motivation of the segment-level reward is quite intuitive, since the token-level reward breaks down the natural semantic unit. The segment-level MDP setting is novel to existing RLHF society, and might inspire more relevant works.

2. The ablation studies are comprehensive.

**Weaknesses:**

1.	The most important procedure in the segment-level RLHF pipeline, semantic segmentation, would increase the computational burden of online RL and is theoretically unguaranteed.

2.	In the experiments, authors train all model on PPO for only one epoch. I wonder whether it is enough for PPO to converge since the paper does not show any training dynamics. This might affect a lot to experimental analysis, since the improvement might stem from faster convergence compared to bandit setting, but with lower final performance.

3.	There are some designs/implementations lack of intuition or analysis. (Question 4/5)

**Questions:**

1.	In Line45, I suggest to remove “/step” from “per-token/step reward signal” since it is easy to connect step-level reward to process reward model rather then token-level reward model in the context.
2.	As mentioned in Line195, different dataset might require different $f(\cdot)$, could you provide any guidance?
3.	Since a new reward normalizer is one of your key contributions, better introduce some corresponding preliminaries in section 2.1
4.	Why does $p$ can be simplest defined as $t/T$? It seems that the latter actions would have a higher p. Maybe provide some intuitive or relevant explanations? The definition of completion portion $p$ would better be clarified.
5.	The enhancement from evenly splitting segment-level rewards to each token is somewhat counter-intuitive and surprising. Can you explain some potential reasons/intuition about this design? Moreover,  decomposing the segment-level reward to token-level signal kind of contradicts to your main novelty, segment-level MDP, and the experiments proves this decomposition is beneficial, which in contrast mitigates the attraction of segment-level MDP setting.
6.	In experiments of RewardBench, how do you determine the final score of a sentence by multiple token-level/segment-level rewards?

---

> ### Author Response · Authors · 2024-11-22
> **Response to Reviewer aY55 (Part 1 / 2)**
>
> We appreciate Reviewer **aY55** for the thorough review.
> We would like to let you know our additional discussions and experiment results in the general response.
> Your concerns are addressed in detail below.
>
>
> > **Q1.**
> Concern on the compute burden and theoretical guarantee of semantic segmentation.
>
> **A.**
> We respectfully note that our semantic segmentation will *not* significantly increase the compute of PPO-based RLHF training. The key is that in policy learning, we use the SFT model for text segmentation (L211). Recall that the PPO objective Eq. (5) itself requires calculating the auto-regressive probability of each sampled token under the SFT model, i.e., $\pi_{\text{SFT}}(y|x) = \prod_{i=1}^{|y|} \pi_{\text{SFT}}(y_i|x, y_{<i})$. This calculation will already calculate the predictive distributions $\pi_{\text{SFT}}(\cdot |x, y_{<i}), \forall i=1, \ldots, |y|$. As a by-product of this calculation of PPO objective, we can efficiently compute the entropy of each predictive distribution $\mathcal{H}(\pi_{\text{SFT}}(\cdot |x, y_{<i}))$ with very few highly-optimized arithmetic operations: $\log$, multiplication, and summation. These entropies are then directly thresholded for text segmentation. To sum up, since *our semantic segmentation procedure re-uses the predictive distributions already calculated in computing the PPO objective*, it incurs no significant compute burden to the PPO-based RLHF training.
>
> We agree with the reviewer that our entropy-based semantic segmentation is more heuristic than rigorous. However, as discussed in L157-161, our method is based on the intuitive assumption that tokens within a semantically complete segment can be more predictable by their generation contexts, while the starting token of a new segment is less so. These assumptions have also been adopted by the literature [e.g., 1,2]. Therefore, we believe our method is valid and viable.
>
> Finally, we also believe that the strong performance of our method validates our segment-level action space for RLHF in LMs, and opens the door for further improvement by adopting/designing more sophisticated semantic segmentation methods.
>
> [1] Li, Bolian, et al. "Cascade reward sampling for efficient decoding-time alignment." arXiv preprint arXiv:2406.16306 (2024).
>
> [2] Malinin, Andrey, and Mark Gales. "Predictive uncertainty estimation via prior networks." Advances in neural information processing systems 31 (2018).
>
> > **Q2.**
> Has PPO converged after one epoch of training?
>
> **A.**
> Thank you so much for this careful thought. Please could you check Appendix G of the newly uploaded revision for evidence on the convergence of PPO training in one epoch. We believe this could further validate our main experimental comparisons.
>
>
> > **Q3.**
> Some designs lack intuition or analysis.
>
> **A.**
> We apologize for any confusion in our design/method description. We will provide detailed reply to your two questions in **Q7** & **Q8** below.
>
> Overall, as one of the earliest works to (re-)consider the granularity of action space in LM’s RLHF, our paper seeks to provide a simple yet effective pipeline to improve reward modeling and PPO-based LM training, by designing a dense yet semantically meaningful action space.  As discussed in the above **Q1**, our competitive results validate our direction and open the door for further improvements by swapping (some of) our designs for more sophisticated ones.
>
> > **Q4.**
> Remove “/step” in L45.
>
> **A.**
> Thank you so much for this careful review! In the newly uploaded revision, we’ve removed it per your suggestion.

---

> ### Author Response · Authors · 2024-11-22
> **Response to Reviewer aY55 (Part 2 / 2)**
>
> > **Q5.**
> Guidance on the selection of $f(\cdot)$ in L195.
>
> **A.**
> As discussed in L183-184, the construction of $f(\cdot)$ factors in our inductive bias on how sequence-level preference is related to each segment’s quality. Apart from using Soft-maximum as $f(\cdot)$ in our main experiments, in the ablation study (Section 4.3 (e)), we further test on using Summation or Average as $f(\cdot)$, i.e., constructing the parametrized sequence evaluation $e_\phi(x,y)$ in Eq. (3) as the summation or average of segment-level rewards in the text sequence. In conjunction with [3], our general guidance for the selection of $f(\cdot)$ is as follows.
>
> 1. Soft-maximum: when chosen responses are distinguished from the rejected ones as containing (a few) key segments (Fig. 2 and L187-188);
> 2. Summation and Average: when chosen responses are identified as having higher average text quality (L527-528), and Average is independent of text-sequence lengths.
> 3. [3] also proposes to use Soft-minimum as $f(\cdot)$ when the chosen responses are identified as not containing “bad words.”
>
> Besides, some RL papers [e.g., 4] also propose to design $f(\cdot)$ as a weighted sum of individual rewards with learnable weights, which may be more data-driven at the cost of more training.
>
> [3] Yang, Shentao, et al. "Preference-grounded token-level guidance for language model fine-tuning." Advances in Neural Information Processing Systems 36 (2024).
>
> [4] Kim, Changyeon, et al. "Preference transformer: Modeling human preferences using transformers for rl." arXiv preprint arXiv:2303.00957 (2023).
>
>
> > **Q6.**
> Add some preliminary on reward normalizers in PPO training.
>
> **A.**
> Thank you so much for this great suggestion! We’ve added a preliminary temporarily in Appendix E of the newly uploaded revision. Please kindly let us know any further suggestions.
>
>
> > **Q7.**
> Explain the completion portion $p$ in L223.
>
> **A.**
> We first respectfully refer the reviewer to Appendix E of the newly uploaded revision for a discussion on why we would like to construct location-aware reward normalizer functions for PPO training. A quick summary is that: the responses’ have a certain linguistic structure, for example, the beginnings of responses are mostly less informational greeting-alike phases. Those text segments thus typically have lower rewards than “key segments” later on in the response. More generally, the responses’ linguistic structure leads to certain correlation between the mean and std of segment-level reward values and the location of segment in the response, e.g., in the early/middle/later part.
>
> With this observation, we would like to construct location-aware reward normalizer functions that can output proper reward normalizers at an arbitrary location of the response (L69-71). The design of $p=t/T$ in L224, i.e., the index of the segment $a_t$ in the response $y$ divided by the total number of segments in $y$, is a simple indicator of the location of $a_t$ in $y$. We apologize for any confusion pertaining to the name “completion portion,” and have changed it to “location indicator” in L221-236 of the newly uploaded revision.
>
> Note that $p=t/T$ can be easily calculated for responses sampled in PPO training, since in PPO training the responses are first *fully* sampled, before calculating the (segment-level) rewards and then normalizing the rewards.
>
> We agree with the reviewer that there could be more sophisticated definitions of the location indicator (“completion portion”) $p$, which opens the door for further improving our method.
>
>
> > **Q8.**
> The intuition of adding even-split reward interpolation and its potential conflict with segment-level MDP.
>
> **A.**
> Thank you so much for this thoughtful question!
>
> In the Appendix F of the newly uploaded revision, we detailedly expand our discussion in Section 4.3 (c) on **(1)** a plausible reason why *no reward interpolation* does not work well; and **(2)** our even-split reward interpolation strategy *does not conflict* with our segment-level design for LM's RLHF. Rather, we regard it as a useful technique to cope with the per-token KL regularization in RLHF PPO training, which is extraneous to RLHF reward modeling/assignment.
>
>
> > **Q9.**
> For RewardBench evaluation, what is the final score from the dense reward models?
>
> **A.**
> As discussed in L302-303, for RewardBench evaluation, we compare the parametrized sequence evaluations $e_\phi(x,y)$ (Eq. (4)) of the two candidate responses, and predict the one with higher $e_\phi(x,y)$ as the chosen.

---

> > ### Comment · Reviewer_aY55 · 2024-11-22
> >
> > Thank you for your detailed response and clarifications.
> >
> > Overall, the author's response has largely addressed my questions.
> > Though their design (e.g.  semantic segmentation, location-aware nomarlizer) are quite heuristic, their segment-level setting is still novel to the society and performant. I will adjust my score accordingly.

---

> > > ### Author Response · Authors · 2024-11-22
> > > **Response to Reviewer aY55**
> > >
> > > Thank you so much for your appreciation of our work and raising your rating. We are delighted that our responses can address your concerns. We will be sure to finalize them in the next version of our manuscript.

---

### Official Review · Reviewer_aW32 · 2024-11-03

**Soundness:** 2
**Presentation:** 2
**Contribution:** 2
**Rating:** 6
**Confidence:** 3

**Summary:**

To retain the optimization benefits of sparse rewards in PPO within RLHF while alleviating potential issues with reward assignment and the anomalies of language intuition, the authors propose a segment-level reward model that assigns rewards to each semantically complete text segment composed of a small number (or just one) of tokens. This approach facilitates RL-based language model training due to denser feedback and provides more accurate training guidance through the semantic completeness of each action. Technically, the authors differentiate the aggregation of segment rewards within the text sequence into parameterized sequence evaluation, addressing the limitations of classical reward normalizers (i.e., the mean and standard deviation of complete sequence rewards) by extending the scalar mean and standard deviation of classical bandit normalizers into functions that can output mean and standard deviation normalizers for rewards at arbitrary positions within the text sequence. They also experimentally test related methods.

**Strengths:**

1. Extensive and detailed ablation experiments have been conducted, providing sufficient evidence for the rationale behind the design choices of the reward model.

2. The use of "segment" instead of "sentence" in the token-level reward modeling process is unique.

**Weaknesses:**

1. The layout of Section 2 is confusing; during the reading process, I did not fully understand the relevance of this paragraph to the experimental setup discussed later.

2. In the experiments presented in Table 6, the segment method does not demonstrate its optimality in most cases. Does this undermine the effectiveness of the method?

**Questions:**

For which specific tasks does the segment method have a greater advantage than the sentence and token methods? The experiments presented in the paper do not seem to highlight the superiority of the segment method.

---

> ### Author Response · Authors · 2024-11-22
> **Response to Reviewer aW32**
>
> We would like to thank Reviewer **aW32** for the careful review.
> We would love to first refer you to our general response for additional discussions and experiments.
> The following are our responses to your concerns.
>
> > **Q1.**
> Confusing layout of Section 2, did not fully understand the relevance of this paragraph to the later experimental setup.
>
>
> **A.**
> We apologize for any confusion in our Section 2. As mentioned in the General Response, in the newly uploaded revision, we fixed the inaccurate descriptions in Section 2.3 pertaining to “completion portion.” We are open to further suggestions on revising our Section 2 and will be sure to implement them in our revised manuscript.
>
> Overall, our Section 2 (Main Method) is meant to be a general method description, while our experimental setup (Section 4.1) provides details on datasets, benchmarks, evaluation metrics, backbone model, and hyperparameters used in our experiments. We are open to any suggestion on how these two sections can be more closely glued together.
>
> > **Q2.**
> Does the less optimal results in Table 6 undermine the effectiveness of the proposed method?
>
> **A.**
> We respectfully note that *Table 6 in Appendix B.1 is **not** our main experiment results/comparisons*. As discussed in L415-417, the purpose of this table is to verify that our method, while achieved strong RLHF training, does not suffer from “alignment tax” [1] where one improves RLHF/alignment metrics but deteriorates general language model abilities (OpenLLM metrics in Table 6).
>
> Since the scope of our paper lies in PPO-based RLHF training, and since we train our LM policy on the preference alignment Ultrafeedback dataset, we do not expect to see our method’s optimality on these OpenLLM metrics. Rather, following the community’s common practice, we keep our eye on any significant degradation of these metrics, which does not appear in our Table 6.
>
> As discussed in L356, our main experiment results are presented in Table 1, where our method shows significant improvement in both reward modeling and PPO-based RLHF training, over both the classical bandit reward approach and the recent token-level reward approach. In Fig. 3, we further visualize the more reasonable reward assignment from our segment-level design of dense reward, compared to the recent token-level design.
>
> [1] Askell, Amanda, et al. "A general language assistant as a laboratory for alignment." arXiv preprint arXiv:2112.00861 (2021).
>
>
> > **Q3.**
> On which specific tasks does the segment-level design outperform the sentence and token design?
>
> **A.**
> We first respectfully refer the reviewer to the above **Q2** for pointers to our main experiment results.
>
> Overall, in this paper, we seek to improve the task of RLHF training by proposing a novel design of RL action space as (semantically complete) text segments, based on intuitions from linguistics and avoiding the sparse reward issue. We believe that our experiment results in Table 1 and Fig. 3 can sufficiently demonstrate our method’s advantages on the specific RLHF tasks of both reward modeling and PPO-based LM training, over both the classical bandit approach and the recent design of token-level action space. Further, Table 1 also presents our method’s advantages over the sentence-level action space (coarser than segment but finer than sequence/bandit).

---

> ### Author Response · Authors · 2024-11-25
> **Follow up with Reviewer aW32**
>
> Dear Reviewer aW32,
>
> We deeply appreciate your time and your thoughtful review!
>
> As the rebuttal period is ending, may we respectfully ask if you still have concerns on our method description (Section 2), the specific tasks where our method tackles, the effectiveness of our method, or have any further concerns?
> We will be more than happy to address your remaining or new concerns during the remaining discussion period.
>
> If our responses have addressed your concerns, would you mind considering re-evaluating our work based on the updated information?

---

> > ### Comment · Reviewer_aW32 · 2024-11-26
> >
> > Thank you for your thoughtful response. I have no further questions, and I will improve my score.

---

> > > ### Author Response · Authors · 2024-11-26
> > > **Thank you for your appreciation**
> > >
> > > Dear Reviewer aW32,
> > >
> > > We are delighted that our response can address all your concerns. We will definitely implement your valuable suggestions into our revised manuscript.

---

### Official Review · Reviewer_KKSu · 2024-11-07

**Soundness:** 3
**Presentation:** 3
**Contribution:** 2
**Rating:** 6
**Confidence:** 4

**Summary:**

This paper tackles the challenge of reward sparsity issue in RLHF by introducing a segment-level reward model. Traditional RLHF approaches generally apply a bandit formulation, assigning rewards only at the sequence level, which leads to sparse feedback. Meanwhile, token-level reward models give more detailed feedback but may be overly granular. This paper proposes a middle-ground solution: segment-level rewards, where rewards are allocated for semantically complete segments within the text. The approach includes dynamic text segmentation, adaptable reward normalization methods, and an interpolation mechanism to further densify training signals. The method is evaluated against popular RLHF benchmarks, demonstrating improvements in reward modeling accuracy and effective LM policy training.

**Strengths:**

The paper conducts extensive experiments with multiple RLHF benchmarks and ablation studies. The proposed method shows competitive performance across all three benchmarks, achieving improvements in both reward model accuracy and downstream LM policy quality. The ablation studies provide thorough insights into certain design choices.

**Weaknesses:**

The proposed method lacks novelty. Densify the fine-grained reward signal for RLHF training have been extensively discussed in previous studies. A key challenge is learning a fine-grained reward model without costly process supervision data. In this paper, the authors directly reference (Yang et al., 2023) in Equation 3 for segment-level reward model learning. Additionally, text sequence segmentation through entropy thresholding in language models has been explored in previous studies as well. It appears that the main contribution of this work lies in showing that segment-level rewards empirically outperform token-level rewards. However, this may not represent a substantial departure from existing ideas.

**Questions:**

1. In Table 1, the segment-level reward model (85.93) outperforms the bandit reward model (81.11), with both models evaluated at the sequence-level on RewardBench. Since the segment-level reward model is also trained using BT loss (Eq 3), could you provide any insights into why the segment-level action space enhances reward modeling?
2. Does using different pre-trained models as reward functions significantly impact the performance?

---

> ### Author Response · Authors · 2024-11-22
> **Response to Reviewer KKSu**
>
> We thank Reviewer **KKSu** for the comprehensive review!
> We would like to first bring to your attention our additional discussions and experiment results in the general response.
> Our responses to your questions are detailed below.
>
> > **Q1.**
> Concern on the novelty of this paper.
>
> **A.**
> We respectfully disagree with the reviewer that our paper lacks novelty.
>
> While densifying the reward signal is currently a popular topic in RLHF research, our paper is among the first to study the important question of *“how dense the fine-grained reward should be?”* We believe that our study is both original and valuable to the research community. Concretely, as illustrated in Figure 3, the popular design of *token-level* dense reward can suffer from reward inconsistency over even a word, which is both counter-intuitive and can result in inferior PPO-based LM training than our segment-level design.
>
> We agree with the reviewer that some techniques to implement/train our segment-level reward model have been individually explored, e.g., in [1,2]. We clearly cited relevant literature in Section 2.2 and consider Section 2.2 as a tactful utilization of existing techniques to realize a more reasonable action space in PPO-based RLHF training.
>
> Apart from reward modeling, we would like to kindly note that in Section 2.3, we propose two techniques for effective PPO training under the dense (segment-level) reward: Location-aware Reward Normalizers via Regression and Within-segment Reward Interpolation. As verified in Section 4.3 (b) & (c), these two techniques are critical to the success of PPO training under the denser reward model. We note that none of our direct references in reward model implementation/training [e.g., 1,2] apply their methods to PPO-based LM training, which turns out requiring some special further designs to utilize dense rewards.
>
> Based on the above reasons, we believe that our paper contains original and meaningful contribution to the field of dense-reward RLHF, which is also acknowledged by Reviewer **aY55**.
>
> [1] Yang, Shentao, et al. "Preference-grounded token-level guidance for language model fine-tuning." Advances in Neural Information Processing Systems 36 (2024).
>
> [2] Li, Bolian, et al. "Cascade reward sampling for efficient decoding-time alignment." arXiv preprint arXiv:2406.16306 (2024).
>
>
> > **Q2.**
> Why does segment-level design enhance reward modeling?
>
> **A.**
> Thank you so much for this insightful question!
>
> In evaluating our segment-level reward model on RewardBench, we compare the parametrized sequence evaluations $e_\phi(x,y)$ (Eq. (4)) of the two candidate responses, and predict the one with higher $e_\phi(x,y)$ as the chosen (L302-303).
>
> As discussed in L183-184, the construction of $e_\phi(x,y)$ factors in our inductive bias on how preference is generated, i.e., the relation between each segment’s quality and the overall sequence’s preferability. Therefore, our better reward model performance, indicating that our constructed $e_\phi(x,y)$ more accurately distinguishes out better responses, may come from our informed model assumption that good responses contain some “key segments” (key phases) to respond to the given prompt (L187-188); and this assumption turns out adequate on RewardBench. For scientific rigor, other datasets/benchmarks may require a different model assumption and hence different construction of $e_\phi(x,y)$ (L195).
>
> Furthermore, our approach of learning the dense reward $r_\phi$ via re-parametrizing the learned sequence-level evaluation (L197-199) could provide a harder learning problem, and thereby help to squeeze out more information from the reward-training dataset into the reward model. As evidence, we did observe that the training of our segment-level reward model converged comparatively slower than the classical bandit reward model, though still to a close final training loss value.
>
>
> > **Q3.**
> Possible to use different pre-trained models as reward functions?
>
>
> **A.**
> We apologize for not fully understanding your questions. We provide our current best answer below.
>
> In General Response **Q3**, we present the experiment result when using Llama3.2-1B-Instruct for the reward model, and Llama3-8B-SFT for the policy. It is clear that our method can train a much stronger larger policy with a smaller reward model, compared to the classical bandit approach.
>
> In general, by our weak requirement of a unified action space in reward modeling and policy training, the rule-of-thumb for using our method for training a larger policy model with a smaller reward model is: both of them and the SFT model should use the same tokenizer, e.g., all belong to the same model series such as Phi 3.1, Llama 3(.x), etc.

---

> > ### Comment · Reviewer_KKSu · 2024-11-25
> >
> > Thank you to the authors for the detailed responses. I believe the current scores accurately reflect my evaluation of the work.

---

> > > ### Author Response · Authors · 2024-11-25
> > > **Response to Reviewer KKSu**
> > >
> > > Dear Reviewer KKSu,
> > >
> > > Thank you so much for the reply. We are delighted that our responses can address your concerns. We will definitely incorporate them in the next version of our manuscript.

---

### Author Response · Authors · 2024-11-22
**General Response**

We deeply appreciate all reviewers for the valuable suggestions. We’ve uploaded a revised manuscript and highlighted the changes in $\text{\textcolor{green}{green}}$. The revisions include:
- Add a preliminary for our design of location-aware reward normalizer functions for PPO training (Appendix E).
- Add a discussion on the necessity and legitimacy of our even-split reward interpolation strategy  (Appendix F)
- Add a verification that PPO training converged after one epoch (Appendix G)
- Fix some inaccurate descriptions in Sections 1 and 2.3.

Below are our additional tabular results of common interests.

 > **Q1:** Use Llama3-8B-SFT [1] as the backbone model.

**A.** We scale up our method to use the popular SFT checkpoint of Llama3-8B from RLHFlow to initialize all relevant models in the training framework. Table A below presents the results of the PPO-trained LM policies.

In Table A, it is clear that our segment-level design again outperforms the strong baselines of bandit/sequence-level action space and the recent token-level design, when using the 8 billion parameter Llama3-8B-SFT as the backbone model. We believe that Table A could further strengthen our main experiment results in Section 4.2.

*Table A: Performance comparison of the PPO-trained LM policies when using Llama3-8B-SFT as the backbone model.*

|  Action Definition | AlpacaEval 2.0 (LC %) | Arena-Hard (WR %) | MT-Bench |
|:------------------:|:---------------------:|:-----------------:|:--------:|
| Raw Llama-3-8B-SFT |         16.31         |        10.4       |   6.82   |
|  Bandit (Sequence) |         20.20         |        18.7       |   7.11   |
|        Token       |         22.14         |        26.0       |   7.13   |
| Segment (**Ours**) |       **23.21**       |      **30.4**     | **7.15** |

[1] https://huggingface.co/RLHFlow/LLaMA3-SFT-v2

 > **Q2:** Use the “RLHF’ed” Phi3-mini-4k-instruct [2] as the backbone model.

**A.**
As per the suggestion of Reviewer **JzqS**, we applied our method onto the “RLHF’ed” (SFT + DPO) model Phi3-mini-4k-instruct, which is used as the model initialization whenever possible. Table B below presents the results of PPO-trained LM policies.

As with our main experiment results, in Table B, our segment-level design again outperforms both the classical bandit formulation and the recent token-level action space, which indicates that our method could be a strong add-on to the existing industrial RLHF pipeline.

*Table B: Performance comparison of the PPO-trained LM policies when using the “RLHF’ed” Phi3-mini-4k-instruct as the backbone model.*

|   Action Definition   | AlpacaEval 2.0 (LC %) | Arena-Hard (WR %) | MT-Bench |
|:---------------------:|:---------------------:|:-----------------:|:--------:|
| Raw Phi-mini-instruct |         18.89         |       25.10       |   7.33   |
|   Bandit (Sequence)   |         20.37         |       26.20       |   7.31   |
|         Token         |         24.01         |       27.92       |   7.37   |
|   Segment (**Ours**)  |       **25.98**       |     **35.80**     | **7.39** |


[2] https://huggingface.co/microsoft/Phi-3-mini-4k-instruct


> **Q3:** A smaller reward model than the policy.

**A.**
To further demonstrate our method’s versatility, as per the suggestion of Reviewer **KKSu**, we use a different backbone for the reward model than the policy. Specifically, we use Llama3.2-1B-Instruct [3] for the reward model, and Llama3-8B-SFT as in the above **Q1** for the policy, SFT, and value function models. Table C below presents the results of the PPO-trained LM policies.

In Table C, it is clear that our segment-level design again outperforms the classical bandit/sequence-level formulation by a significant margin. We believe that this experiment could demonstrate our method’s versatility in the practical setting where a smaller reward model is used to train a larger policy.

*Table C: Performance comparison of the PPO-trained LM policies when using Llama3.2-1B-Instruct for the reward model, and Llama3-8B-SFT for the policy.*

|  Action Definition | AlpacaEval 2.0 (LC %) | Arena-Hard (WR %) | MT-Bench |
|:------------------:|:---------------------:|:-----------------:|:--------:|
| Raw Llama-3-8B-SFT |         16.31         |        10.4       |   6.82   |
|  Bandit (Sequence) |         19.70         |        14.5       |   7.10   |
| Segment (**Ours**) |       **23.54**       |      **29.2**     | **7.20** |

[3] https://huggingface.co/meta-llama/Llama-3.2-1B-Instruct

---

### Meta-Review · Area_Chair_TcM8 · 2024-12-18

**Metareview:**

This paper proposes a new paradigm for RLHF, by training and utilizing a segment-level reward model (in the middle of token-level and sequence-level reward models) to capture more subtle (but not overly subtle) semantics. While the reviewers generally found the motivation to be clear and interesting, they also raised concerns regarding
* The novelty of the work (e.g., segment-level reward has been studied in Wu et al. "Fine-Grained Human Feedback Gives Better Rewards for Language Model Training" but the authors didn't discuss it).
* The empirical effectiveness (the evaluation setup that the authors constructed, including the base model choice and hyperparameter settings, made it hard to compare against prior work).
Overall, the paper is promising but could benefit significantly from another round of revisions for a resubmission.

**Additional Comments On Reviewer Discussion:**

The reviewers didn't reach a consensus in the discussion, with some reviewers expressing their unaddressed concerns. The positive reviewers also acknowledged the weaknesses of the paper and were not willing to support the paper strongly.

---

### Decision · Program_Chairs · 2025-01-22

Reject